# TRACING CONCEPT CIRCUITS TO AUDIT AND STEER VISION TRANSFORMERS

## ABSTRACT

Advanced vision models, *e.g.,* Vision Transformers (ViTs), might base their decisions on spurious cues, even for correct predictions. To ensure their safe deployment in high-stakes applications, it is essential to *audit* ViT decision-making processes and *steer* them away from unsafe predictions. Traditional interpretation methods typically attribute predictions to salient pixels or neurons. However, such simplified correlations often overlook the concepts encoded in internal representations, which can be the true causes of failures. To this end, we develop an interpretation toolbox, *ViSAE*, to trace the *concept circuits* from ViT representations. These circuits enable users to *(i) audit* models by identifying spurious shortcuts, and *(ii) steer* model behaviors by amplifying or suppressing specific concepts along influential paths. Specifically, we construct a *neuroscience-*motivated probing suite (64K images and 16K concepts) that mirrors the human visual cortex hierarchy. Building upon the data, we train Sparse Autoencoders (SAEs) to read concepts directly from the representations of ViT and trace their causal relationships. Extensive experiments and ablation studies show that our probing suite outperforms existing counterparts by 20× in concept coverage efficiency and 28.7% in interpretation accuracy. We demonstrate that using *ViSAE*, we can identify spurious decision paths, localize concepts on pixels, and diagnose the model failure modes. Furthermore, our toolbox enables model steering by editing concepts within representations, which improves worst-group accuracy on the WaterBirds dataset by 48.2%. Our data, code, and models are in
`https://anonymous.4open.science/r/ViSAE-7405`.

## 1 INTRODUCTION

Machine Learning (ML) models might base their decisions on spurious features, even for correct predictions (Arjovsky et al., 2019; Sagawa et al., 2019). This fragility can yield severe consequences in high-stakes applications, such as medical diagnosis (Ong Ly et al., 2024; Brown et al., 2023) and autonomous driving (Magnussen et al., 2020; Danks & London). To diagnose ML models, traditional interpretable ML (IML) methods, as shown in Fig. 1, typically attribute the predictions to salient pixels (Selvaraju et al., 2017; Lundberg & Lee, 2017) or neurons (Ghorbani & Zou, 2020; Bau et al., 2017). However, such correlations provide limited insight into the model's inner workings (Olah et al., 2020). This leads us to a critical question: Can we *audit* the internal mechanisms of ML models and, when needed, *steer* models away from spurious correlations?

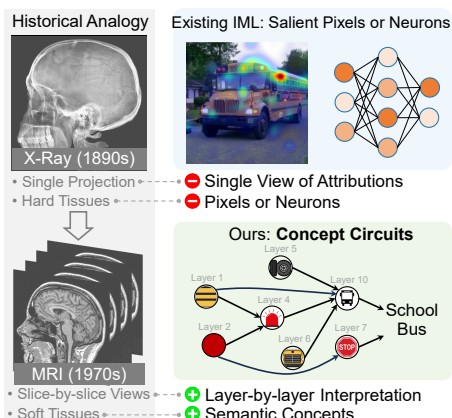

Figure 1: A historical analogy comparing traditional IML to X-rays, and our proposed approach to MRI.

Uncovering ML model internal mechanisms is non-trivial. One major challenge is the phenomenon of *superposition* (Elhage et al., 2022), where neurons encode multiple unrelated concepts to maximize parameter efficiency. This results in *polysemantic representations* that are uninterpretable to humans. Recent work in Sparse Autoencoders

(SAEs) (Bricken et al., 2023; Huben et al., 2023; Zou et al., 2023) shed light on addressing such polysemanticity in language transformers (*e.g.,* GPT-2 (Radford et al., 2019)). They typically learn a sparse code over a dictionary of basis features to reconstruct polysemantic representations, encouraging each feature to capture a *monosemantic concept* (Olshausen & Field, 1997; Ng et al., 2011). However, adapting SAEs to vision transformers (ViTs) (Dosovitskiy et al., 2020) misses critical *infrastructures:* **(1) What probing data enables faithful interpretation of ViT inner workings?** Recent studies (Hindupur et al., 2025) show that what SAEs can see and explain is shaped by the data they are trained on and interpret. However, existing vision SAE works (Rao et al., 2024; Lim et al., 2024; Stevens et al., 2025; Thasarathan et al., 2025) often rely on off-the-shelf image sets (*e.g.,* ImageNet (Deng et al., 2009)) and generic text vocabularies (Oikarinen & Weng; Bhalla et al., 2024). Consequently, their interpretations are typically skewed toward object-level concepts. **(2) How to represent the inner workings of ViT in a human-readable way?** Existing vision SAE works typically *extract discrete concepts from a single layer,* often the final embeddings. As a result, there is no causal guarantee that these concepts drive final predictions. Although circuit discovery methods (Olah et al., 2020; Conmy et al., 2023) can trace information flow over neurons or attention heads, the resulting circuits are computational graphs of low-level components. Such graphs are difficult for humans to parse and to map to high-level reasoning.

Building such infrastructures is non-trivial. We develop *ViSAE* interpretation toolbox to fill the gaps:

**(1) Data: Neuroscience-motivated probing suite.** To mitigate data bias, we build a new probing suite, including images and concepts, following the hierarchy of the human visual system (Goodale & Milner, 1992; Carandini et al., 2005). Concretely, the *human visual cortex* processes information across four abstraction levels: Primitive, Intermediate, Object, and Scene. *On the image side,* we collect images from seven vision datasets whose major content maps to the four abstraction levels. After redundancy reduction, our probing set comprises 64K images to maximize concept coverage and training efficiency. *On the concept side,* to enable auto-interpretation and evaluation, we leverage GPT-5 to annotate each image with fine-grained concepts across the four abstraction levels, resulting in 16K unique concepts (*e.g.,* "stripes", "skimming"). As a result, our image set outperforms the popular ImageNet baseline by 20× in concept coverage efficiency; the same SAEs interpreted on our concepts outperform existing concept sets by 28.7%.

**(2) Algorithm: Concept circuit tracing.** To reveal how concepts evolve in the model, we propose a two-step tracing algorithm. *Top-down concept reading:* Building upon our probing data, we train SAEs to decompose representations at each transformer layer into sets of basis features. We then use the vision-language embedding space of CLIP (Radford et al., 2021) to map the learned SAE features to human concepts. *Bottom-up causal tracing:* We trace the concept circuits across layers by counterfactual edits. Specifically, we set the SAE activation of a concept in the earlier layer to zero and reconstruct the representation via the SAE decoder. We measure the resulting change in the SAE activation of the target concept in the later layer as the causal influence (*i.e.,* indirect effects). By repeating this process across layers and concepts, we construct a directed, weighted graph where nodes are concepts and edges represent causal influences. Our interpretations are faithful to model predictions, outperforming existing counterparts by 33.7% in downstream steering tasks.

**(3) Applications: Auditing and steering.** Our *ViSAE* enables diagnostic auditing and corrective steering of ViTs. *For auditing,* it supports users to trace internal information flow, localize concepts in pixel space, and diagnose model failure modes. More importantly, *for steering,* it offers a set of conceptual "knobs" to control model behavior by editing concepts within representations. For example, by turning down spurious correlated concepts (*e.g.,* land backgrounds), *ViSAE* improves the worst-group accuracy in the WaterBirds (Sagawa et al., 2019) dataset by 48.2%.

Contributions: Our novelty is *not* another SAE variant. Instead, we introduce a toolbox for interpreting the inner workings of ViTs, *i.e.,* a faithful infrastructure that is largely missing in the existing literature. Our toolbox consists of: (1) A neuroscience-motivated probing suite (64K images, 16K concepts) for SAE training and auto-interpretation. (2) A two-step causal tracing algorithm for holistic, layer-wise discovery of concept circuits within ViTs. (3) Extensive empirical validation, including SAE benchmarking and applications in representation auditing and steering.

## 2 METHOD

In this section, we first review the basics of SAEs (Sec. 2.1), then introduce the construction of our probing suite (Sec. 2.2), followed by how it is used to train SAEs and to trace concept circuits (Sec. 2.3), and finally how to use the toolbox to audit and steer ViTs (Sec. 2.4). Overview in Fig. 3.

### 2.1 PRELIMINARY

Sparse Autoencoders (SAEs) (Ng et al., 2011; Bricken et al., 2023) are proposed to interpret polysemantic neurons by formulating it as a *sparse dictionary learning* problem (Olshausen & Field, 1997). The objective is to learn an *overcomplete* set of sparse, disentangled basis features (*i.e.,* concepts) that can reconstruct the input data through linear combination (Thasarathan et al., 2025). Concretely, SAE consists of an encoder that expands the input dimensionality, namely $f : \mathbb{R}^d \to \mathbb{R}^m$, $m > d$, and a decoder $g : \mathbb{R}^m \to \mathbb{R}^d$. A vanilla ReLU-SAE (Bricken et al., 2023) is given by:

$$\mathbf{h} = f(\mathbf{x}) = \mathrm{ReLU}(\mathbf{W}_{\mathrm{enc}}\mathbf{x} + \mathbf{b}_{\mathrm{enc}}), \quad \hat{\mathbf{x}} = g(\mathbf{h}) = \mathbf{W}_{\mathrm{dec}}\mathbf{h} + \mathbf{b}_{\mathrm{dec}}, \tag{1}$$

where $\mathbf{W}_{\mathrm{enc}}, \mathbf{W}_{\mathrm{dec}}^\top \in \mathbb{R}^{m \times d}$ and $\mathbf{b}_{\mathrm{enc}}, \mathbf{b}_{\mathrm{dec}} \in \mathbb{R}^m$. Note that in the decoder parameter matrix $\mathbf{W}_{\mathrm{dec}}$, each column $\mathbf{w}_i$ represents a learned basis feature, namely $\hat{\mathbf{x}} = \sum_{i=0}^{m-1} h_i \mathbf{w}_i + \mathbf{b}_{\mathrm{dec}}$. The training objective minimizes the reconstruction error while enforcing sparsity in the latent code:

$$\mathcal{L}(\mathbf{x}) = ||\mathbf{x} - \hat{\mathbf{x}}||_2^2 + \lambda ||\mathbf{h}||_1. \tag{2}$$

Given the SAE, two factors largely affect interpretation quality. **(1) Quality of the input representation x.** The data distribution (*e.g.,* images) that produces $\mathbf{x}$ governs what features are even learnable. As noted in Sec. 1, object-centric datasets bias learning toward specific abstraction levels (Stevens et al., 2025; Thasarathan et al., 2025). **(2) Interpretation method of SAE features.** Existing works typically label a feature $\mathbf{w}_i$ by inspecting its top-activating images (Thasarathan et al., 2025; Pach et al., 2025), but this process is subjective and hard to scale. We address these challenges in the following sections with a new probing suite and an automated interpretation method.

### 2.2 NEUROSCIENCE-MOTIVATED PROBING SUITE

As discussed in Sec. 1, for SAE training, object-centric datasets often provide limited concept coverage for the full spectrum of visual processing. To fill this gap, we construct a probing suite that offers broad concept coverage motivated by the hierarchical organization of the human visual cortex from neuroscience (Goodale & Milner, 1992; Carandini et al., 2005; DiCarlo et al., 2012).

**Background: visual cortex hierarchy.** The human visual system processes information along abstraction levels. As shown in Fig. 2: (1) At the *primitive level*, the primary visual cortex encodes basic visual primitives, *e.g.,* colors, edges, and curves. (2) At the *intermediate level*, secondary visual cortex integrate these primitives into more complex patterns *e.g.,* textures, materials, and geometric shapes. (3) At the *object level*, these patterns are combined into identifiable entities like tables, airplanes, or animals, supporting object recognition in the temporal lobe. (4) Finally, at the *scene level*, higher-order regions represent actions, spatial relations, and interactions, enabling reasoning about context and events, often associated with the parietal lobe.

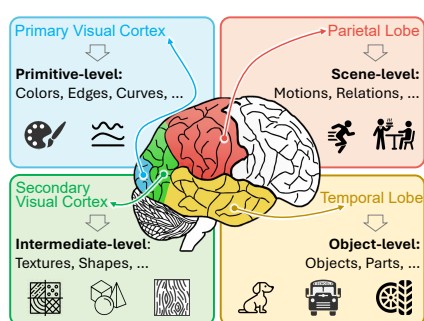

Figure 2: Our data curation is motivated by the hierarchy of human visual cortex.

**Neuroscience-motivated probing image set.** To maximize the concept coverage of SAE, we first collect probing images from seven vision datasets mirroring the hierarchy above. Specifically, (1) at the *primitive level,* we collect images from the DTD (Cimpoi et al., 2014) and Broden (Bau et al., 2017); (2) at the *intermediate level*, we collect images from Broden and ShapeNet (Chang et al., 2015); (3) at the *object level*, we collect images from ImageNet (Deng et al., 2009) and Visual Genome (Krishna et al., 2017); and (4) at the *scene level*, we collect images from Place365 (Zhou et al., 2017) and MSCOCO (Lin et al., 2014). However, naively aggregating images reduces SAE training efficiency while offering minimal concept coverage benefits. This is because repeated views

Figure 3: Overview of our *ViSAE* toolbox for interpreting ViT inner workings. ***Left:*** Motivated by the human visual cortex hierarchy, we construct a probing suite (64K images + 16K concepts) for SAE training and interpretation. ***Middle:*** Our top-down concept reading and bottom-up concept circuit tracing algorithms. ***Right:*** Our mechanistic view of ViT inner workings enables various downstream applications, such as concept localization, failure mode analysis, and model steering.

Table 1: Comparison of concept coverage (%) across probing image sets. We obtain the concept coverage of images by calculating the CLIP embedding similarity between images and our ground truth concepts (details in **Appendix B**). As shown, our probing image set demonstrates superior concept coverage across all levels of visual abstraction, with over $20\times$ higher coverage efficiency.

| Probing Image Set | Data Source | # of Images | Concepts Covered by Images (%) | | | | | Coverage Efficiency ↑ (%/1K Images) |
|---|---|---|---|---|---|---|---|---|
| | | | Primitive | Intermediate | Object | Scene | Avg. | |
| ImageNet | ImageNet | 1,281K | 81.0 | 78.2 | **97.7** | 59.0 | 78.9 | 0.06 |
| MSCOCO | MSCOCO | 118K | 69.6 | 65.4 | 80.4 | **63.1** | 69.6 | 0.59 |
| Ours | Primitive Level: DTD, Broden; Intermediate Level: Broden, ShapeNet; Object Level: ImageNet, VisualGenome; Scene Level: Place365, MSCOCO; | 64K | **87.1** | **80.6** | 92.6 | 61.7 | **80.5** | **1.26** |

cause SAEs to waste limited model capacity on the same high-frequency concepts, biasing the learned features. To address this issue, we prune the initial pool (121K raw image candidates) by removing one image from every pair with a cosine similarity greater than 0.85 in the *CLIP-ViT-B-32* embedding space. As a result, our final set contains 64K probing images. Tab. 1 shows our superior concept coverage, outperforming the popular ImageNet baseline by $20\times$ in coverage efficiency.

**Neuroscience-motivated concept set.** To enable the proposed automatic interpretation (Sec. 2.3) of SAE features, a candidate pool of concepts with strong visual grounding is required. Existing vocabularies are typically mined from text,

Table 2: Concept set statistics and human evaluation results.

| Level | # of Concepts |
|---|---|
| Primitive | 1,073 |
| Intermediate | 1,723 |
| Object | 10,534 |
| Scene | 2,720 |
| Total | 16,050 |

| Evaluator | Faithfulness | Completeness |
|---|---|---|
| Human_A | $4.65 \pm 0.06$ | $4.72 \pm 0.04$ |
| Human_B | $4.83 \pm 0.20$ | $4.73 \pm 0.06$ |
| Human_C | $4.90 \pm 0.05$ | $4.78 \pm 0.18$ |
| Avg. | 4.79 | 4.74 |

*e.g.,* frequent n-grams from Google Books (Oikarinen & Weng) or LAION captions (Bhalla et al., 2024), which are typically skewed toward linguistically frequent terms and drift from the images. To reduce this bias, we generate concepts *from the images themselves*. Concretely, for each image in our probing set, we use GPT-5 (OpenAI, 2025) to annotate present concepts under the same four-level hierarchy. In practice, we fix the generation hyperparameters (*e.g.,* temperature=1.0, seed=42) and conduct prompt engineering to ensure the consistency of concept annotation (see our full prompts in **Appendix D**). As shown in Tab. 2, the resulting concept set contains 16K unique one- and two-gram concepts, and each probing image is paired with a set of ground truth concepts that are presented on it.

**Human Evaluations.** To ensure the quality of the concept annotations, we further conduct human evaluations. We evaluate two metrics: Faithfulness, which measures whether concepts are visually grounded in the image, and Comprehensiveness, which measures whether they cover all visual abstraction levels. We randomly sample three groups of images ($3\times50$) paired with our concept an-

notations and hire graduate students to evaluate all different groups in a 0-5 Likert score. As shown in Tab. 2, our concepts are of high quality and the annotation process is consistent across images, as evidenced by an average Likert rating above 4.7/5.

## 2.3 CONCEPT CIRCUIT TRACING ALGORITHM

**Top-down concept reading.** Although SAEs decompose polysemantic representations into disentangled features, interpreting the semantics of these features remains challenging. Existing methods typically retrieve top-activating samples and summarize them into a specific concept (Bills et al., 2023; Pach et al., 2025). However, for vision models, this process relies on subjective human review and does not scale. To address this issue, we introduce an automated method to "read" concepts directly from representations. Specifically, by leveraging the aligned embedding space of vision-language models, we map each SAE feature to the most semantically aligned textual concept in our concept set. Let $\mathbf{W}_{\text{dec}}$ be the decoder weight matrix of a trained SAE, where each column $\mathbf{w}_i$ is a basis feature. Using our probing image set $\mathcal{D}_{\text{probe}} = \{x_1, ..., x_N\}$, we extract the feature activation vector $q_i = \left[ h_i(x_1), h_i(x_2), \ldots, h_i(x_N) \right]^\top \in \mathbb{R}^N$ for neuron $i$ over all images. For the concept set $\mathcal{D}_{\text{concept}} = \{c_1, ..., c_M\}$, we compute a concept activation matrix $P \in \mathbb{R}^{N \times M}$ using a VLM, *e.g.,* CLIP (Radford et al., 2021), where $P_{nm}$ is the embedding similarity between image $x_n$ and concept $c_m$. We associate SAE feature $i$ with concept $c_m$ using the Soft Weighted Point-wise Mutual Information (Soft-WPMI) (Oikarinen & Weng) score:

$$\text{Sim}(i, c_m) = \log \mathbb{E}_{x \sim \mathcal{D}_{\text{probe}}}[\alpha_i(x) \cdot P_{xm}] - \lambda \log p(c_m), \tag{3}$$

where $\alpha_i(x_n) = \frac{\exp(q_i[n])}{\sum_{j=1}^N \exp(q_i[j])}$ is the softmax-normalized activation of neuron $i$, $p(c_m) = \frac{1}{N} \sum_{n=1}^N P_{nm}$ is the marginal prevalence of concept $c_m$, and $\lambda > 0$ controls the penalty for overly frequent concepts. The final concept label for SAE feature $i$ is determined by:

$$c^*(i) = \arg \max_{c_m \in \mathcal{D}_{\text{concept}}} \text{Sim}(i, c_m). \tag{4}$$

**Bottom-up causal tracing.** Now we mapped SAE features to concepts. However, how these discrete concepts relate to one another and how they compose into the final decision remains unclear. To address this, we trace the causal influence among concepts and toward the prediction from bottom up. Specifically, inspired by *activation patching* methods (Meng et al., 2022; Conmy et al., 2023), we define edges by quantifying the causal importance via counterfactual interventions. Let $\alpha_j^t$ be the activation of a target concept $c_j^t$ in a downstream target layer $t$. Consider a concept $c_i^s$ extracted from source layer $s$ via SAE, with activation $\alpha_i^s$. To measure its influence on a target concept $c_j^t$, we construct two layer $s$ representations of the same input $x$: the original representation $r_{\text{clean}}$ and a *patched* representation $r_{\text{patch}}$, in which $c_i^s$ is ablated by setting its activation $\alpha_i^s$ to zero and reconstructing the representation via the SAE decoder. In this case, we define the causal influence of $c_i^s$ on $c_j^t$ by measuring the *indirect effect* (IE) (Pearl, 2001):

$$\text{IE}_{i \to j}^{s \to t} = \text{IE}\left(\alpha_j^t; c_i^s; r_{\text{clean}}, r_{\text{patch}}\right) = \alpha_j^t(r_{\text{clean}}) - \alpha_j^t(r_{\text{clean}} \,|\, \text{do}(\alpha_i^s = \alpha_i^s(r_{\text{patch}}))) . \tag{5}$$

We can also obtain the contribution of concept $c_i^s$ to the final prediction $y$ by:

$$\text{IE}_{i \to y}^s = \text{IE}(y; c_i^s; r_{\text{clean}}, r_{\text{patch}}) = y(x_{\text{clean}}) - y(r_{\text{clean}} \,|\, \text{do}(\alpha_i^s = \alpha_i^s(r_{\text{patch}}))) . \tag{6}$$

By repeating this procedure for all concepts over all layers, we obtain a directed graph where nodes represent concepts and edges are weighted by the causal importance of the target node regarding the final prediction. Concretely, the weight of an edge from $c_i^s$ to $c_j^t$ is given by $\text{IE}_{i \to j}^{s \to t} \cdot \text{IE}_{j \to y}^t$. We model the ViT forward pass as a deterministic structural causal model (SCM), and our concept nodes refine this into a more fine-grained SCM on top of it. Consequently, all edges are directed and acyclic, *i.e.,* $s \to t$ only when $t > s$. This *concept circuit* reveals how primitive features are progressively composed into intermediate patterns and, ultimately, high-level semantics that drive the model's predictions.

## 2.4 APPLICATIONS: AUDITING AND STEERING

Building upon the concept circuits identified, our *ViSAE* toolbox offers practical toolkits for model analysis and intervention.

**Auditing.** *(1) Trace information flow:* Users can visualize the concept circuit for arbitrary image, revealing the pathways of causal influence from low-level primitives to the final prediction (Fig. 5). *(2) Localize concepts on pixels:* Concretely, to localize $c_i$ on images, we calculate the cosine similarities between $\mathbf{w}_i$ and each image token $t_j \in \mathbb{R}^d$ from the corresponding transformer layer. This generates a saliency map $h_i = \frac{\langle \mathbf{w}_i, t_j \rangle}{\|\mathbf{w}_i\|_2 \, \|t_j\|_2}$ that highlights the regions where the concept $c_i$ is most strongly activated (Fig. 6). *(3) Diagnose failure modes:* By comparing concept circuit differences between correct and wrong prediction groups, users can systematically analyze the failure modes, identifying if errors stem from spurious features or missing critical factors (Fig. 7).

**Steering.** Our *ViSAE* enables targeted control of model behavior by concept editing. *(1) Suppress spurious concepts:* Mitigate shortcut learning by setting the activation of an undesired concept to zero, effectively removing its influence from the computation graph (Fig. 8). *(2) Amplify robust concepts:* Enhance the effect of desirable features by manually increasing their activation (Tab. 5).

## 3 EXPERIMENTS

In this section, we first benchmark different SAE variants on image data (Sec. 3.1), then evaluate their interpretation accuracy (Sec. 3.2). We next demonstrate how to leverage our toolbox to audit model behavior (Sec. 3.3) and to steer the model toward desired predictions (Sec. 3.4).

### 3.1 BENCHMARKING SAES ON IMAGE DATA

We systematically evaluate SAE variants on disentangling and reconstructing vision representations.

**Settings:** We evaluate representative SAEs, *i.e.,* ReLU-SAE (Bricken et al., 2023), BatchTopK-SAE (Bussmann et al., 2024), Matryoshka-SAE (Bussmann et al., 2025), Gated-SAE (Rajamanoharan et al., 2024a), JumpReLU-SAE (Rajamanoharan et al., 2024b). For each SAE architecture, we train multiple variants with five expansion factors ($2\times$, $4\times$, $8\times$, $16\times$ and $32\times$) and five average $L_0$ sparsities ($8$, $16$, $32$, $64$ and $128$), resulting in 25 instances in total. The *expansion factor* refers to the expansion ratio between the SAE latent dimension and input dimension. We train SAEs on the representations (CLS and image token separately, $2\times12$ SAEs in total) extracted from the residual stream of each layer in ViT (*CLIP-ViT-B-32* unless noted), using our probing image set as input. For each layer, we train SAEs separately on *cls* tokens and *image* tokens. Details of these SAE architectures are in **Appendix E**.

**Metrics:** *(1) $L_0$ Sparsity*: the average number of non-zero activations. *(2) Reconstruction Error (RE)*: the mean squared error (MSE) between the input representation and the SAE reconstructed representation. *(3) Decoder Orthogonality (DO)* (Zaigrajew et al., 2025): the mean cosine similarity between each pair of decoder columns. This metric measures whether an SAE encodes concepts with distinct semantic meanings. *(4) Dead Neuron (DN)*: the proportion of SAE basis features remaining consistently inactive (zero activation) across the whole training dataset. *(5) Monosemanticity (MS)* (Pach et al., 2025): whether each basis feature of an SAE consistently activates on images of the same semantics. Details of the metrics are in **Appendix F.**

**Results:** We observe that all SAE architectures share similar trends with respect to $L_0$ sparsity on all five evaluation metrics. To balance the reconstruction quality and monosemanticity, we set sparsity of 128 and expansion factors $8\times$ in practice. Fig. 4 shows the benchmark results for expansion factors $8\times$. Implementation details and full tables are in **Appendix G & H.1.**

### 3.2 EVALUATIONS OF INTERPRETATION ACCURACY

We develop a new metric based on our probing suite to evaluate the interpretation accuracy of SAE training data and concept sets for auto-interpretation.

**Settings:** Similar to Sec. 3.1, We train SAEs on the representations (CLS and image tokens separately) extracted from the residual stream of each layer in *CLIP-ViT-B-32*. (1) We compare different training data (downsampled to 60K), including our probing images, ImageNet (Deng et al., 2009), and MSCOCO (Lin et al., 2014). (2) We compare using different concept sets to interpret the same SAEs, including our concept set ($\sim$16K), LAION frequent words (15K) (Bhalla et al., 2024), and Google Books common English words (20K) (Oikarinen & Weng). (3) We compare using existing

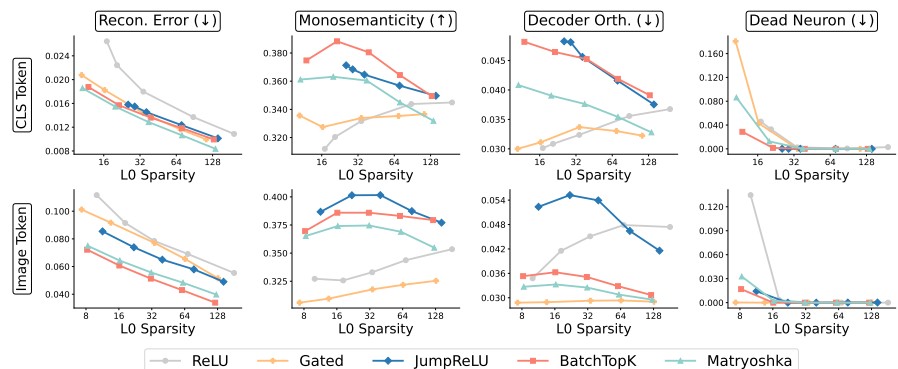

Figure 4: Benchmark results for expansion factor ($8\times$). As shown, BatchTopK-SAE strikes a better trade-off across all metrics on image data. Therefore, in subsequent experiments we use the BatchTopK-SAE with expansion factor $= 8\times$ and $L_0$ Sparsity $= 128$. Full tables in **Appendix H.1.**

fine-grained interpretability datasets to train and interpret SAEs with our probing suite, including Broden (Bau et al., 2017) and LaBo (Yang et al., 2023). Note that LaBo only provides concept sets, so here we use ImageNet to train the SAEs and use LaBo's concepts for ImageNet to interpret the SAE features (Sec 2.3).

Table 3: Comparison of Interpretation Accuracy.

| Probing Image Set | Concept Set | Interpretation Accuracy (%) | | |
|---|---|---|---|---|
| | | Top-10 | Top-20 | Top-30 |
| Ours-64K | Google-20K | 10.7 | 14.4 | 16.7 |
| Ours-64K | LAION-15K | 17.3 | 22.2 | 25.4 |
| ImageNet | Ours-16K | 32.2 | 43.5 | 50.7 |
| MSCOCO | Ours-16K | 34.9 | 45.0 | 51.2 |
| Broden | Broden | 16.8 | 23.9 | 26.8 |
| ImageNet | LaBo-ImageNet | 15.9 | 18.4 | 20.3 |
| Ours-64K | Ours-16K | **36.6** | **47.6** | **54.1** |

**Metrics:** We split the probing images into train and test sets (60K/4K). Since ground-truth concepts are available for our test images (Sec. 2.2), we calculate *interpretation accuracy* by measuring the fraction of ground-truth concepts covered within the top-$K$ concepts read by SAEs from all layers. We use CLIP-based semantic match rather than a raw string match to mitigate vocabulary circularity.

**Results:** As shown in Tab. 3, in the top-30 extracted concepts, SAEs trained on our probing images consistently outperform existing datasets by 2.9% and 3.4%. Moreover, for the same SAEs, using our concept set consistently outperforms existing concept sets by 28.7% and 37.4%. Our probing suite outperforms existing fine-grained interpretability datasets by 27.3%.

### 3.3 AUDITING

In this section, we demonstrate how to use our toolbox to audit the model behaviors. Specifically, *ViSAE* enables concept circuit tracing, concept localization, and model failure mode diagnosis.

**Trace concept circuits.** Fig. 5 shows the concept circuit examples traced by our method. Beyond faithfully identifying decision pathways, the circuits reveal a layer-wise progression that is similar to the human visual system: early layers detect low-level primitives (colors, textures), while deeper layers compose these cues into higher-level semantics (objects, relations/motion).

**Localize concepts on pixels.** Qualitatively, Figs. 5 & 6 show examples of our concept localizations. Our method can accurately localize concepts across visual abstraction levels. Note that we do not manually choose the layer; the SAE activations determine it. For example, if an image strongly activates an SAE feature labeled "wooden texture" in its layer-3 representation, we use that layer-3 feature for localization. Quantitatively, as shown in Tab. 4, our heatmaps on the Quantus (Hedström et al., 2023) benchmark improve over the existing attribution-based method (Chefer et al., 2021) by 3.7% using the VOC2007 dataset in terms of localization accuracy.

Table 4: comparison of heatmap localization accuracy.

| Method | Point Game | Attribution Localization |
|---|---|---|
| Chefer et al. | 41.9 | 32.3 |
| Ours | **45.0** | **36.0** |

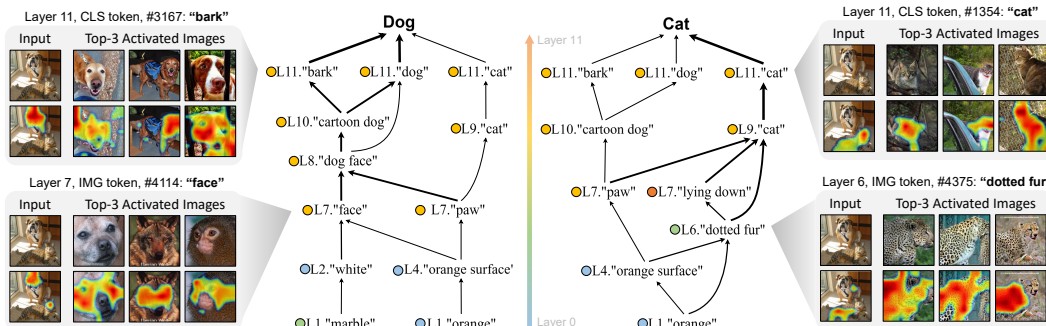

Figure 5: Visualization of concept circuits. For an input image containing both a dog and a cat, our method traces the unique causal pathways leading to each prediction. The circuit for "dog" composes primitive and intermediate concepts (*e.g.,* "orange" and "marble") into high-level semantics (*e.g.,* "bark" and "dog"). In contrast, the circuit for "cat" relies on a different set of concepts (*e.g.,* "dotted fur"). The results show that our method can faithfully audit the inner workings of the ViT and highlight the responsible concepts. More examples in **Appendix H.2.**

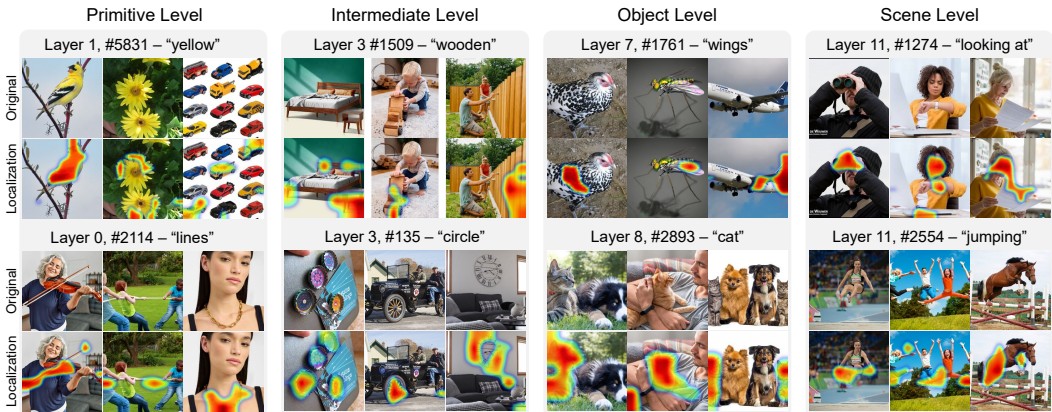

Figure 6: Localize concepts in the pixel space. Notably, our method can even localize highly abstract semantics, such as "looking at", by highlighting both the subject (*i.e.,* the person) and the object involved (*i.e.,* the paper). More examples in **Appendix H.2.**

**Diagnose failure modes.** Beyond instance-level auditing, understanding the failure patterns of the model at a global level is essential for improving its robustness. To this end, we demonstrate how to leverage our *ViSAE* to diagnose the failure modes of the *CLIP-ViT-B-32* model on the ImageNet validation set. Specifically, we identify all 38 classes for which more than 40% of the images are consistently misclassified into the same incorrect class. For each such class, we use our trained SAEs to extract the most frequently activated con-

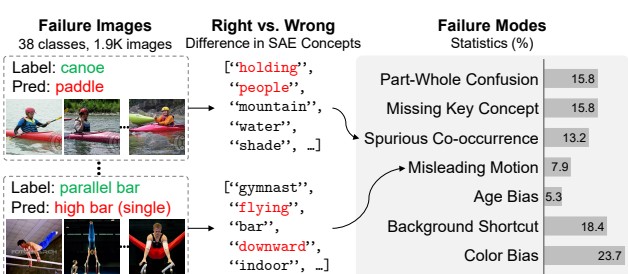

Figure 7: Failure mode analysis. We identify seven failure modes of CLIP on the ImageNet-val set. For example, CLIP tends to misclassify parallel bar images as high bars when the gymnast is "flying" or oriented "downward".

cepts from both correctly and incorrectly classified images. By comparing these two concept groups, we summarize key semantic differences, eventually organizing them into seven distinct failure modes (Fig. 7). Building on these analyses, our *ViSAE* can be extended to support data quality control by identifying mislabeled, ambiguous, or systematically biased samples.

Table 5: Model steering accuracy on the WaterBird dataset. The Worst Group is referring to "Waterbird on Land", where the land background is the spurious factor and we intervene on such information to steer the model predictions.

| Method | Steer Spuri. Corr. | Overall Acc. (%) | Worst Group Acc. (%) | Δ |
|---|---|---|---|---|
| CBM | None | - | 37.3 | - |
| | Remove | - | 51.8 | + 14.5 |
| SpLiCE | None | - | 48.0 | - |
| | Remove | - | 60.0 | + 12.0 |
| DN-CBM | None | - | 57.5 | - |
| | Remove | - | 71.3 | + 13.8 |
| PCBM | None | - | 50.3 | - |
| | Remove | - | 74.7 | + 24.4 |
| Ours | None | 79.7 | 50.3 | - |
| | Enhance | 74.5 | 5.3 | − 45.0 |
| | Remove | 85.2 | **98.5** | + **48.2** |

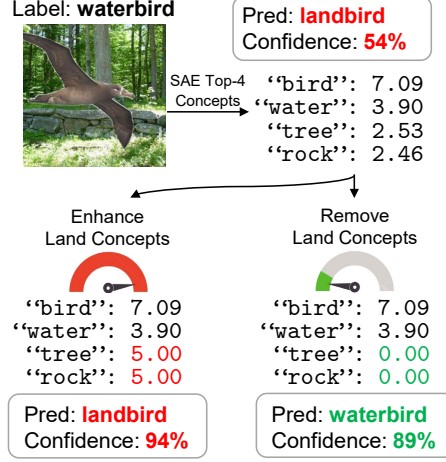

Figure 8: Qualitative examples of steering on WaterBird dataset.

## 3.4 STEERING

In this section, we demonstrate the capability of our *ViSAE* in steering model behaviors. Specifically, we use the SAEs to intervene on internal representations by selectively removing or enhancing specific concepts, thereby steering the final predictions in a controlled and interpretable manner.

**Settings.** We use the WaterBirds dataset (Sagawa et al., 2019) to evaluate robustness to spurious correlations between bird species (land/water) and their backgrounds. The training set amplifies this spurious correlation (*e.g.,* waterbirds on water), while a 5% "worst-group" in the test set breaks it (*e.g.,* waterbirds on land). A linear classifier trained on *CLIP-ViT-B-32*'s final-layer CLS tokens achieves only 50.3% accuracy on this worst group, confirming heavy reliance on background cues. We compare with Concept Bottleneck Models (CBM) (Koh et al., 2020), SpLiCE (Bhalla et al., 2024), DN-CBM (Rao et al., 2024), and Post-hoc CBM (PCBM) (Yuksekgonul et al., 2022) that can steer the model.

Table 6: Ablation study on the impact of probing image sets for model steering.

| SAE Probing Set | Steer Spuri. Corr. | Worst Group Acc. (%) |
|---|---|---|
| None | None | 50.3 |
| ImageNet | Remove | 63.9 |
| MSCOCO | Remove | 95.4 |
| Ours | Remove | **98.5** |

**Edit spurious concepts.** We mitigate spurious correlations by using our SAE to ablate background-related concepts (e.g., "grass", "land") in the worst-group samples. This is done by setting their activations to zero, reconstructing new CLS tokens, and re-classifying. As shown in Tab. 5, this intervention boosts worst-group accuracy by 48.2%. Conversely, enhancing these concepts degrades performance by 45.0%, demonstrating precise bidirectional control via our concept-level "knobs".

**Ablation study on different probing image sets.** We further conduct ablation studies to evaluate the influence of the probing image set for training SAEs on steering performance. To ensure a fair comparison, we keep the SAE architecture and all training hyperparameters fixed, and train SAEs using ImageNet, MSCOCO, and our curated probing image set, respectively. As shown in Tab. 6, the SAEs trained on our probing image set significantly outperforms its counterparts, even though those also achieve improvements over existing methods.

## 4 RELATED WORK

In this section, we will discuss the most related works. A more detailed version is in **Appendix A.**

**Interpretable Machine Learning (IML).** Existing IML methods typically attribute model predictions to either input features or model components. These are broadly categorized into two paradigms. *Post-hoc methods*, such as GradCAM (Selvaraju et al., 2017), LIME (Ribeiro et al.,

2016), and SHAP (Lundberg & Lee, 2017), typically provide an saliency map of input pixels; *Intrinsic methods,* such as ProtoPNet (Chen et al., 2019) and explanation-guided learning (Ross et al., 2017), incorporate interpretability directly into the model architecture. However, the former is often limited to input-output correlations, and the latter relies on custom architectures that are not easily generalizable across tasks (Adebayo et al., 2018; Rudin, 2019). Differently, our method interprets internal representations post hoc, without requiring architectural changes to the explainee model.

**Concept-based interpretability.** Concept-based methods address the limited intelligibility of saliency maps by explaining predictions via human-understandable concepts. Techniques include TCAV (Kim et al., 2018) (testing sensitivity to predefined concepts), Network Dissection (Bau et al., 2017) (labeling neurons with visual concepts), and Concept Bottleneck Models (Koh et al., 2020) (architecturally enforcing concept predictions). However, these approaches rely on predefined concept sets or annotations, limiting their scalability in open-world settings (Yuksekgonul et al., 2022; Margeloiu et al., 2021). Different from existing works, we use SAEs to "read" concepts directly from model representations without supervised concept labels.

**Mechanistic Interpretation (MI).** MI methods aim to reverse-engineer the internal mechanisms of deep models (Nanda et al., 2023; Bereska & Gavves, 2024). *Bottom-up* approaches, such as the Circuits framework (Olah et al., 2020; Conmy et al., 2023), dissect neural connectivity but yield low-level graphs that lack human interpretability (Marks et al., 2024). *Top-down* methods like Sparse Autoencoders (SAEs) (Olshausen & Field, 1997; Ng et al., 2011) learn disentangled features to address superposition. In language models, SAEs are trained on large corpora (e.g., The Pile (Gao et al., 2020), Gemma (Team et al., 2024)) and interpreted via LLM summarization (Bills et al., 2023). For vision, however, SAEs face two key challenges: biased datasets (e.g., ImageNet (Deng et al., 2009)) limit concept coverage, and feature interpretation remains subjective (Thasarathan et al., 2025; Pach et al., 2025). Our toolbox bridges these gaps by curating data and auto-interpretation.

## 5 CONCLUSION

In this paper, we introduced *ViSAE*, a comprehensive interpretation toolbox for auditing and steering ViTs by tracing human-understandable concept circuits. Our approach bridges the gaps in mechanistic interpretability for vision models by constructing a neuroscience-motivated probing suite that provides broad coverage of visual concepts. By training SAEs on our data and leveraging a two-step causal tracing algorithm, we can reveal the circuits of concepts layer-by-layer, transforming opaque model representations into interpretable causal graphs. Furthermore, we show that *ViSAE* is not only a powerful auditing tool for identifying spurious correlations and failure modes but also enables effective model steering through concept-level interventions.

**Limitations.** Despite its promising results, our approach has limitations. Although our auto-interpretation achieves high coverage in extracting ground-truth concepts, its resolution is constrained by the granularity of the concept set and embedding space of VLM (*e.g.,* CLIP). We observe instances where different SAE features are mapped to the same concept label. However, qualitative analysis by human reviewers, inspecting top-activating images and concept localizations, reveals that these features often capture visually and semantically nuanced sub-concepts. This indicates that the SAEs learn even finer-grained visual concepts than our current auto-interpretation pipeline.

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

APPENDIX

## A RELATED WORK

**Interpretable Machine Learning (IML).** IML methods aim to uncover the reasons behind model predictions and are typically categorized as: Post-hoc methods, such as GradCAM (Selvaraju et al., 2017), LIME (Ribeiro et al., 2016), and SHAP (Lundberg & Lee, 2017), provide explanations by attributing predictions to input features; Intrinsic methods, such as ProtoPNet (Chen et al., 2019) and explanation-guided learning (Ross et al., 2017), incorporate interpretability directly into the model architecture by design. However, the former is often limited to surface-level input-output relationships, while the latter depends on custom architectures that are not easily generalizable across tasks (Adebayo et al., 2018; Rudin, 2019). In contrast, our method decouples interpretation from prediction and instead analyzes internal representations post hoc, without requiring architectural changes to the explainee model.

**Concept-based interpretability.** Concept-based methods emerged as a response to the limitations of attribution methods, where saliency maps often fail to provide human-interpretable explanations (Adebayo et al., 2018; Kindermans et al., 2019). TCAV (Kim et al., 2018) uses curated probing sets to evaluate a model's sensitivity to predefined concept directions. Network Dissection (Bau et al., 2017) assigns semantics to individual neurons using human-annotated labels. ACE automatically discovers salient concept clusters in latent space. Concept Bottleneck Models (CBMs) (Koh et al., 2020) enforce a human-defined concept layer within the network, enabling transparency and intervention. However, these methods typically require concept annotations or assume a closed-world setting with a fixed concept vocabulary, making them struggle to scale up to open-world concept discovery that does not assume the set of concepts is a known prior (Yuksekgonul et al., 2022; Margeloiu et al., 2021). Our method differs by directly extracting concepts from pretrained models without requiring concept supervision or architecture changes.

**Mechanistic Interpretation (MI).** MI methods (Nanda et al., 2023; Bereska & Gavves, 2024) aim to uncover the internal computational mechanisms of deep models and have shown promising progress, particularly in language models. Bottom-up approaches, such as the Circuits framework (Olah et al., 2020; Conmy et al., 2023), dissect individual neurons and their connectivity to reveal functional subcomputations. However, the resulting units (*e.g.,* neurons) are often not interpretable to humans (Marks et al., 2024). Top-down approaches, including representation engineering (Zou et al., 2023) and Sparse Autoencoders (SAEs) (Olshausen & Field, 1997; Ng et al., 2011), address this by learning disentangled, monosemantic features that map more naturally to human-understandable concepts, mitigating the feature superposition issue. While SAEs decompose polysemantic representations into monosemantic features, ensuring comprehensive coverage and interpreting their semantic meanings remains challenging. For language models, existing approaches (Huben et al., 2023; Lieberum et al., 2024) typically train SAEs on massive text corpora, such as the Pile ($\sim$7M) (Gao et al., 2020) or Gemma ($\sim$3T) (Team et al., 2024), to broaden concept coverage, and interpret features by prompting LLMs (*e.g.*, GPT-4 (Achiam et al., 2023)) to summarize the semantics of top-activating examples (Bills et al., 2023). For vision models, however, available datasets are usually biased toward object-level concepts (*e.g.,* ImageNet (Deng et al., 2009)), might not cover the full spectrum of visual processing. Furthermore, the top-activating images of SAE features often show ambiguous semantics, making their interpretation subjective (Thasarathan et al., 2025; Pach et al., 2025).

## B CONCEPT COVERAGE CALCULATION

To measure concept coverage in a dataset-agnostic manner, we leverage a Top Percentile Method that evaluates how well each concept is represented by the most similar images in the dataset. For each concept $c_i$, we compute the cosine similarity between the concept's text embedding and all image embeddings (CLIP-ViT-B-32) in the dataset, yielding a similarity vector $\mathbf{s}_i \in \mathbb{R}^N$ where $N$ is the dataset size. Rather than relying on maximum similarity (which can be noisy) or overall mean similarity (which may be dominated by irrelevant images), we calculate the mean similarity of the top $k$ most similar images, where $k = \max\left(1, \lceil N \cdot p/100 \rceil\right)$ and $p$ is a small percentile (typically

0.005%), namely:

$$\text{Coverage Score }(c_i) = \frac{1}{k} \sum_{j=1}^{k} s_{i,\text{top}-j}, \qquad (7)$$

where $s_{i,\text{top}-j}$ represents the $j$-th highest similarity score between concept $c_i$ and the images in the dataset. A concept is considered "covered" at threshold $\tau$ if Coverage Score $(c_i) \geq \tau$. In practice, we set $\tau = 0.25$ as a meaningful similarity between modalities.

This approach is robust to dataset size variations and provides a stable measure of concept representation quality by focusing on the images that most strongly exhibit each concept, while avoiding the influence of outliers or the vast majority of irrelevant images.

## C  CONCEPT COUNT CALCULATION

To understand the semantic distribution of existing concept sets across different abstraction levels, we perform a match analysis that assigns each of their concept to its best-matching abstraction level in our ground truth taxonomy. Given an existing concept set $\mathcal{C} = \{c_1, c_2, ..., c_N\}$ and ground truth concepts organized by abstraction levels $\mathcal{G}_{\text{gt}} = \{\mathcal{G}_{\text{primitive}}, \mathcal{G}_{\text{intermediate}}, \mathcal{G}_{\text{object}}, \mathcal{G}_{\text{scene}}\}$, we first compute the semantic similarity between each new concept and all ground truth concepts $\text{sim}(c_i, g_i)$ using CLIP text embeddings (CLIP-ViT-B-32). We then identify the best-matching ground truth concept for each new concept:

$$g_i^* = \arg\max_{g_j \in \bigcup_l g_l} \text{sim}(c_i, g_j) \qquad (8)$$

A concept $c_i$ is considered well-matched if its maximum similarity exceeds a threshold $\tau$ (0.9 in practice). Each well-matched concept is then assigned to the abstraction level of its best-matching ground truth concept.

This analysis reveals the semantic composition of existing concept sets, showing how many concepts align with each abstraction level (primitive, intermediate, object, scene) and identifying concepts that may not represent visual concepts.

## D  FULL PROMPT

---

**Our Prompt to Obtain Concept Annotations**

```
Analyze the provided image and identify common visual concepts
grouped into these categories: Color, Edge, Texture, Shape,
Object, Part, Motion, and Relation.

Category Specifications:
- Color: Only identify the basic color patterns present in the
image, avoid specifying hues or shades (e.g., "red", "bright
yellow").
- Edge: Detect prominent visual boundaries, lines, or edge
characteristics (e.g., "soft edge", "hard contour", "straight
edge").
- Texture: Recognize surface qualities or repeated patterns,
avoid referencing associated objects (e.g., "dotted", "smooth",
"wooden", "woven", "grid").
- Shape: Note geometric or organic forms, avoid referencing
associated objects (e.g., "round", "triangular", "cylindrical").
- Object: List distinct items or entities visible within the
image (e.g., "car", "tree", "bottle").
- Part: Specify individual segments or components of objects
(e.g., "wheel", "leaf", "handle").
- Motion: Indicate observed or implied actions or movement
(e.g., "running", "driving", "falling").
```

---

```
   – Relation: Describe positional or functional relationships
   between objects or parts. The concepts should always be
   prepositions or verbs, avoid including any nouns (e.g., "on
   top", "next to", "contains").

   Instructions:
   – For each category, extract relevant visual concepts, limiting
   each concept to a maximum of three words.
   – Output your results as a JSON object. Each category ("Color",
   "Edge", "Texture", "Shape", "Object", "Part", "Motion",
   "Relation") must be present as a key.
   – Each key's value should be an array of strings, where each
   string is a concept in that category.
   – If no concepts are detected in a category, return an empty
   array for that key.
   – Include only the specified categories and fields; no extra
   data or metadata.

   # Output Format
   {
     "Color": [string],
     "Edge": [string],
     "Texture": [string],
     "Shape": [string],
     "Object": [string],
     "Part": [string],
     "Motion": [string],
     "Relation": [string]
   }

   Example Output:
   ```json
   {
     "Color": ["red", "bright yellow"],
     "Edge": ["soft edge"],
     "Texture": ["rough", "smooth"],
     "Shape": ["round"],
     "Object": ["car", "tree"],
     "Part": ["wheel", "leaf"],
     "Motion": ["driving"],
     "Relation": ["on top", "next to", "inside"]
   }
   ```
```

# E DETAILS ON SAE ARCHITECTURES

In this section, we provide details for the SAE variants used in our study. These variants differ primarily in how they induce sparsity in the hidden representation $\mathbf{h}$.

**ReLU SAE** (Bricken et al., 2023). This is the standard sparse autoencoder using ReLU nonlinearity followed by an $L_1$ sparsity penalty. Given the input representation $\mathbf{x} \in \mathbb{R}^{d_{\text{in}}}$, encoder, decoder weights $\mathbf{W}_{\text{enc}}, \mathbf{W}_{\text{dec}}^{\top} \in \mathbb{R}^{d_{\text{hid}} \times d_{\text{in}}}$ and biases $\mathbf{b}_{\text{enc}}, \mathbf{b}_{\text{dnc}} \in \mathbb{R}^{d_{\text{hid}}}$, the hidden representation $\mathbf{h} \in \mathbb{R}^{d_{\text{hid}}}$ is given by:

$$\mathbf{h} = \text{ReLU}(\mathbf{W}_{\text{enc}}\mathbf{x} + \mathbf{b}_{\text{enc}}), \tag{9}$$

and the reconstruction is:

$$\hat{\mathbf{x}} = \mathbf{W}_{\text{dec}}\mathbf{h} + \mathbf{b}_{\text{dec}}. \tag{10}$$

The ReLU SAE is trained to minimize a loss function:

$$\mathcal{L} = \mathcal{L}_{\text{reconstruction}} + \mathcal{L}_{\text{sparsity}} = \|\mathbf{x} - \hat{\mathbf{x}}\|_2^2 + \lambda\|\mathbf{h}\|_1, \tag{11}$$

where $\lambda$ is the regularization coefficient to control sparsity. This variant is simple and effective, but sparsity is indirectly controlled by $\lambda$.

**BatchTopK SAE** (Bussmann et al., 2024). Instead of applying a soft sparsity penalty, this variant enforces hard sparsity by retaining only the top-$k$ activations across a batch of $n$ samples. Specifically, it retains the $n \times k$ largest activations across the batch and zeros out all others:

$$\mathbf{h} = \text{BatchTopK}(\mathbf{W}_{\text{enc}}\mathbf{x} + \mathbf{b}_{\text{enc}}). \tag{12}$$

The loss becomes:

$$\mathcal{L} = \|\mathbf{x} - \mathbf{W}_{\text{dec}}\mathbf{h} + \mathbf{b}_{\text{dec}}\|_2^2. \tag{13}$$

This provides deterministic sparsity and is particularly suitable for interpretability-focused applications.

**Matryoshka SAE** (Bussmann et al., 2025). Matryoshka SAE is inspired by the idea of nested sparsity levels. It produces multiple nested representations $\{\mathbf{h}^{(1)}, \ldots, \mathbf{h}^{(K)}\}$ such that each $\mathbf{h}^{(k)}$ satisfies $\mathbf{h}^{(1)} \subseteq \ldots \subseteq \mathbf{h}^{(K)}$, where $\mathbf{h}^{(K)} = \text{BatchTopK}(\mathbf{W}_{\text{enc}}\mathbf{x} + \mathbf{b}_{\text{enc}})$. The reconstruction loss is computed over all levels:

$$\mathcal{L} = \sum_{k=1}^{K} \|\mathbf{x} - \mathbf{W}_{\text{dec}}\mathbf{h}^{(k)} + \mathbf{b}_{\text{dec}}\|_2^2. \tag{14}$$

This encourages a hierarchical structure in the learned features and facilitates interpretability at multiple granularity levels.

**JumpReLU SAE** (Rajamanoharan et al., 2024b). JumpReLU replaces the standard ReLU with a modified activation function that enforces a minimum activation threshold:

$$\text{JumpReLU}(z) = \begin{cases} z, & z > \tau \\ 0, & \text{otherwise} \end{cases}, \tag{15}$$

where $\tau$ is a fixed threshold (e.g., $\tau = 0.001$). This nonlinearity encourages fewer active units by cutting off low activations more aggressively than ReLU, resulting in sparser codes even without explicit sparsity penalties.

**Gated SAE** (Rajamanoharan et al., 2024a). Gated SAE uses multiplicative gating to modulate activations. Each hidden unit has a learned gate $g_i \in [0, 1]$, typically computed via a sigmoid:

$$\mathbf{h}_i = \sigma(\mathbf{a}_i^\top \mathbf{x}) \cdot \text{ReLU}(\mathbf{w}_i^\top \mathbf{x} + b_i). \tag{16}$$

This allows the model to selectively suppress irrelevant features and provides an adaptive mechanism to control sparsity, potentially improving both interpretability and flexibility.

## F EVALUATION METRICS

**Reconstruction Error.** This metric quantifies how well the autoencoder reconstructs the input data from the sparse code. Formally, given input $\mathbf{x} \in \mathbb{R}^{d_{\text{in}}}$ and reconstructed output $\hat{\mathbf{x}} = \mathbf{W}_{\text{dec}}\mathbf{h} + \mathbf{b}_{\text{dec}}$, the Reconstruction Error is defined as:

$$\text{Reconstruction Error} = \mathbb{E}_{\mathbf{x} \sim \mathcal{D}} \left[ \|\mathbf{x} - \hat{\mathbf{x}}\|_2^2 \right]. \tag{17}$$

Lower Reconstruction Error indicates that the SAE preserves more input information. However, extremely low reconstruction error may come at the cost of losing sparsity or interpretability.

**Monosemanticity** (Pach et al., 2025). Monosemanticity is a measure of how consistently a neuron responds to semantically similar inputs. Intuitively, a neuron is considered *monosemantic* if its highest activations occur for a group of inputs that are semantically coherent (e.g., images depicting the same object or concept). Specifically, it measures the visual similarity between the top-$k$ activated inputs for each neuron.

Formally, let $f(\mathbf{x}) = \mathbf{h} \in \mathbb{R}^{d_{\mathrm{hid}}}$ be the encoder output for input $\mathbf{x}$, and let $h_i$ denote the activation of the $i$-th neuron. For each neuron $i \in \{1, \ldots, d_{\mathrm{hid}}\}$, we identify the top-$k$ inputs from the dataset $\mathcal{D}$ that elicit the highest activations:

$$\mathcal{T}_i = \mathrm{TopK}\left(\{(\mathbf{x}, h_i(\mathbf{x})) \mid \mathbf{x} \in \mathcal{D}\}\right). \tag{18}$$

We then compute the average pairwise cosine similarity between the embeddings of the top-$k$ input images. Let $\phi(\mathbf{x}) \in \mathbb{R}^d$ be a feature embedding of image $\mathbf{x}$ obtained from the CLIP ViT-B/32 model. The monosemanticity score for neuron $i$ is:

$$\mathrm{Mono}(i) = \frac{2}{k(k-1)} \sum_{1 \leq p < q \leq k} \frac{\phi(\mathbf{x}_p)^\top \phi(\mathbf{x}_q)}{\|\phi(\mathbf{x}_p)\|_2 \cdot \|\phi(\mathbf{x}_q)\|_2}, \quad \text{where } \{\mathbf{x}_1, \ldots, \mathbf{x}_k\} = \mathcal{T}_i. \tag{19}$$

Finally, the overall monosemanticity score for the autoencoder is obtained by averaging over all neurons:

$$\mathrm{Monosemanticity} = \frac{1}{d_{\mathrm{hid}}} \sum_{i=1}^{d_{\mathrm{hid}}} \mathrm{Mono}(i). \tag{20}$$

Higher values indicate that neurons respond selectively to visually similar inputs, which supports more interpretable and disentangled representations.

**Decoder Orthogonality** (Zaigrajew et al., 2025). To enhance interpretability, it is desirable that the learned basis features (*i.e.*, decoder columns) are disentangled. One way to promote this is to encourage orthogonality among decoder vectors. Let $\mathbf{W}_{\mathrm{dec}} = [\mathbf{w}_1, \ldots, \mathbf{w}_{d_{\mathrm{hid}}}] \in \mathbb{R}^{d_{\mathrm{in}} \times d_{\mathrm{hid}}}$ denote the decoder weight matrix. The Decoder Orthogonality is defined as the mean pair-wise cosine similarity between each pair of decoder columns:

$$\mathrm{Decoder\ Orthogonality} = \frac{2}{d_{\mathrm{hid}}(d_{\mathrm{hid}} - 1)} \sum_{1 \leq i < j \leq d_{\mathrm{hid}}} \mathbf{w}_i^\top \mathbf{w}_j. \tag{21}$$

A smaller value implies greater orthogonality and lower redundancy among the learned features. Perfect orthogonality occurs when all decoder vectors are mutually orthogonal unit vectors.

**Dead Neuron.** This metric measures the fraction of hidden units that are never activated across a dataset. A hidden neuron is considered "dead" if its activation is zero for all inputs in a dataset $\mathcal{D}$. Let $\mathbf{h}(\mathbf{x})$ be the hidden representation on input $\mathbf{x}$ and $h_i(\mathbf{x})$ the $i$-th dimension. Define the dead neuron set:

$$\mathcal{D}_{\mathrm{dead}} = \left\{ i \in \{1, \ldots, d_{\mathrm{hid}}\} \ \middle| \ \sum_{\mathbf{x} \sim \mathcal{D}} h_i(\mathbf{x}) = 0 \right\}. \tag{22}$$

The Dead Neuron ratio is:

$$\mathrm{Dead\ Neuron} = \frac{|\mathcal{D}_{\mathrm{dead}}|}{d_{\mathrm{hid}}}. \tag{23}$$

A high dead neuron ratio indicates underutilization of the model capacity, which may suggest over-regularization or poor feature allocation. On the other hand, a moderate level of dead neurons may naturally emerge in highly sparse encoders.

## G  IMPLEMENTATION DETAILS

**Training Details.** We train all Sparse Autoencoders (SAEs) on our probing image set using the *cls* tokens and *image* tokens from the residual stream of each layer in the CLIP ViT-B/32 model (each layer has two SAEs). Each SAE consists of an overcomplete linear encoder and a sparse decoder, with the decoder columns constrained to unit $\ell_2$ norm. For benchmark experiment, we vary the expansion factor $ef \in \{2, 4, 8, 16, 32\}$, defined as $d_{\mathrm{hid}} = ef \cdot d_{\mathrm{in}}$, and $L_0$ sparsity $L_0 \in \{8, 16, 32, 64, 128\}$ for all SAE architectures.

**Optimization and Scheduler.** We train all models using a modified Adam optimizer that enforces unit-norm constraints on decoder columns. We use a fixed batch size of 4096, learning rate $\eta = 3 \times 10^{-4}$, and train for 100 epochs. No learning rate decay or warmup is applied.

**Hyperparameter Choice.** We control sparsity in BatchTopK and Matryoshka SAEs by directly setting the top-$k$ values $k \in 8, 16, 32, 64, 128$. For ReLU, JumpReLU, and Gated SAEs, we perform

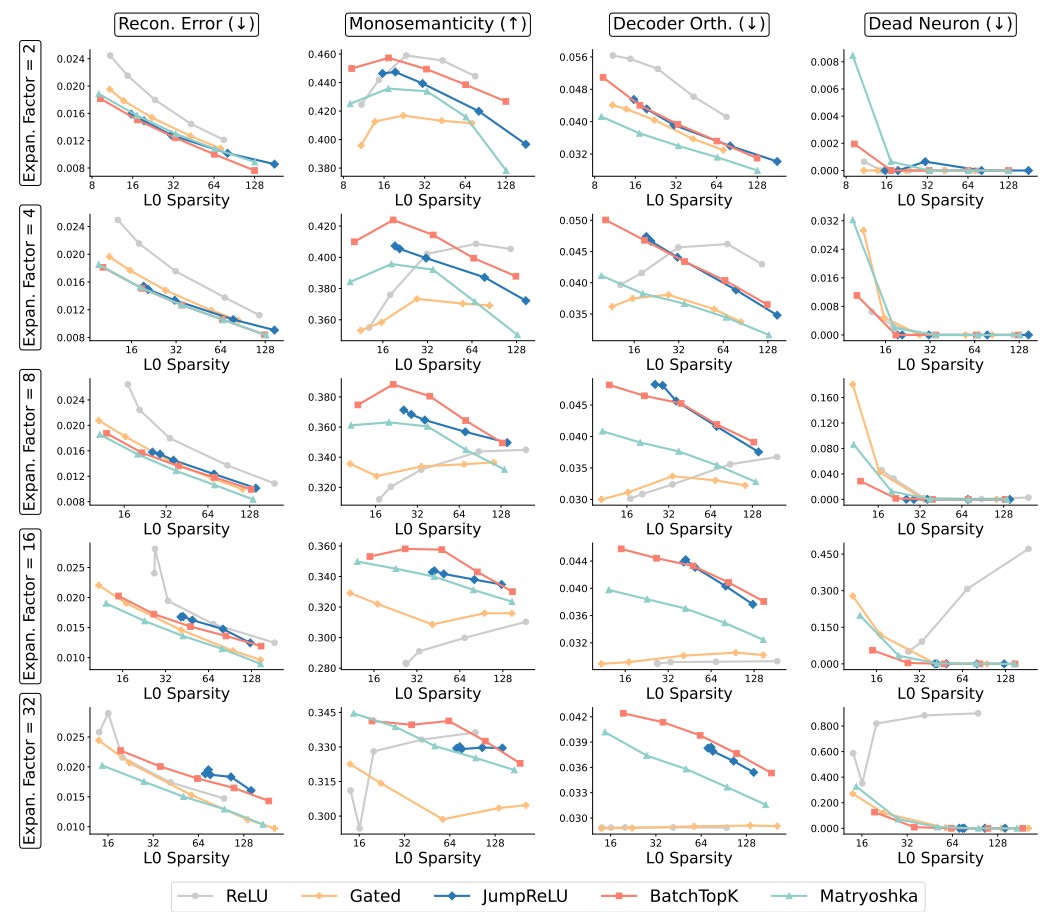

Figure 9: Full Benchmark results for *cls* tokens.

a sweep over sparsity regularization strength $\lambda$ to approximate the target $L_0$ sparsity levels. In Matryoshka SAEs, we use nested hidden representations with cumulative fractions $\frac{1}{32}, \frac{1}{16}, \frac{1}{8}, \frac{1}{4}, \frac{1}{2}, 1$. For JumpReLU SAEs, we fix the jump threshold $\tau = 0.001$ throughout all experiments. For the Monosemanticity metric, we compute pair-wise similarity between $k = 9$ top activated images for each feature basis, for the Interpretation Accuracy metric, we retrieve $k = 3$ concepts from the concept set for each feature basis.

## H ADDITIONAL EXPERIMENTAL RESULTS

### H.1 FULL BENCHMARK RESULTS

We provide full benchmark results in Figs. 9 & 10.

### H.2 ADDITIONAL CONCEPT CIRCUIT AND LOCALIZATION EXAMPLES

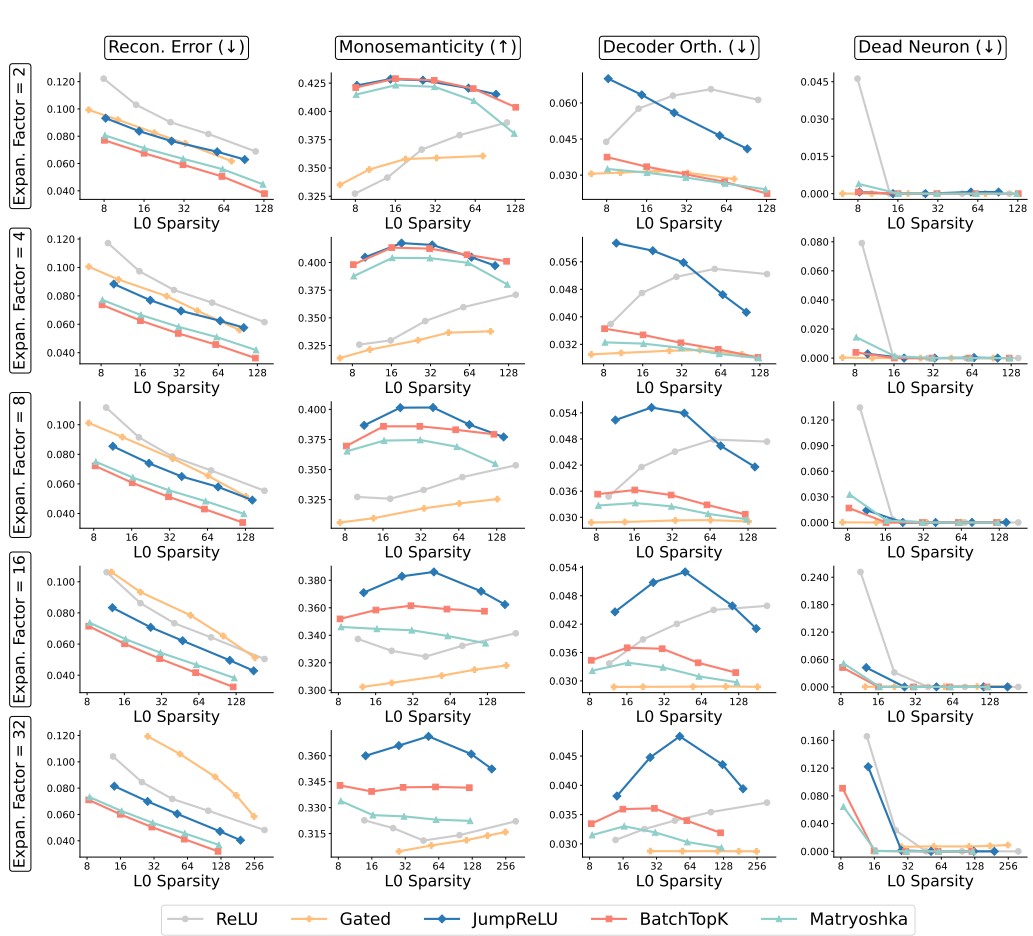

Figure 10: Full Benchmark results for image tokens.

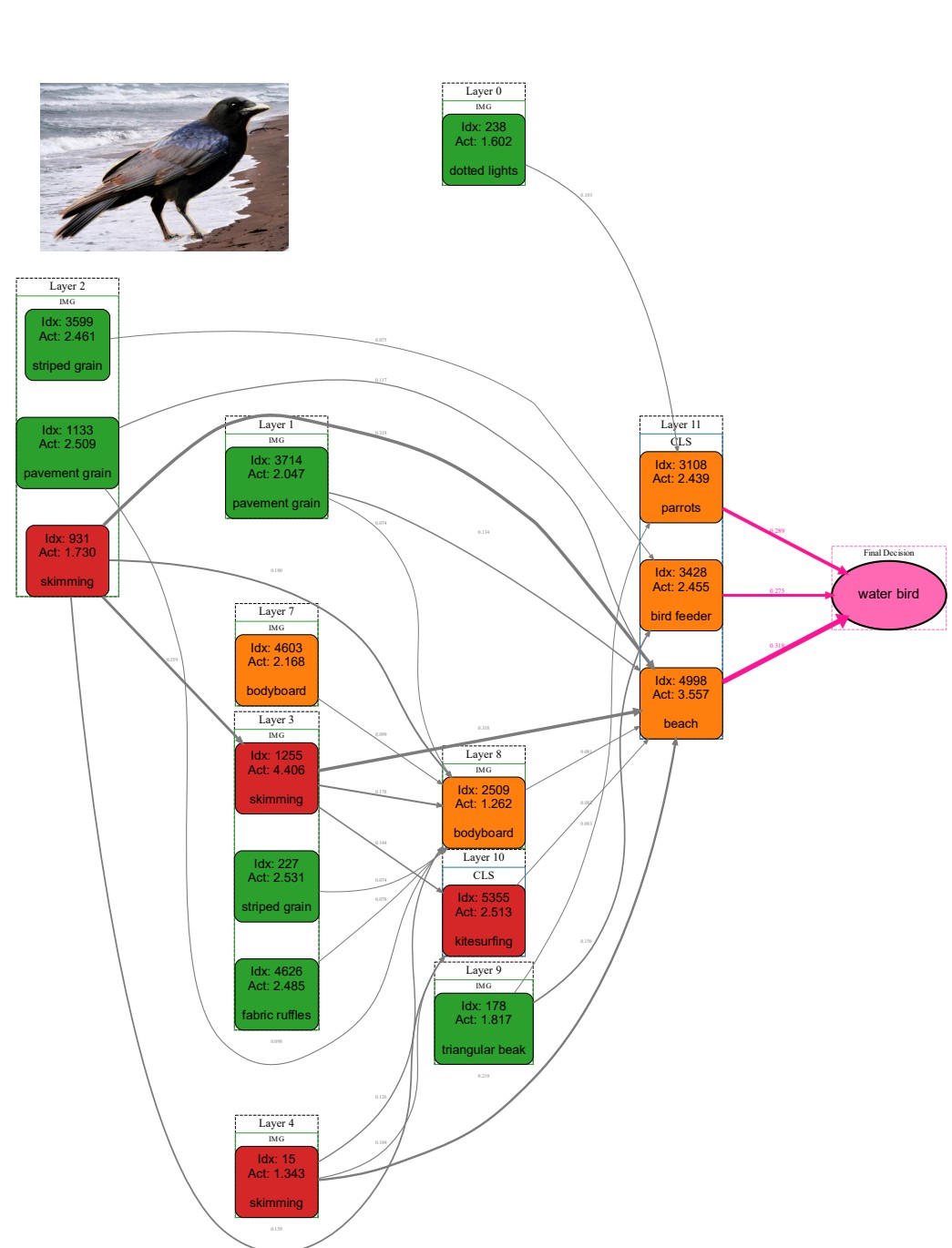

Figure 11: Concept circuit for landbird on water background.

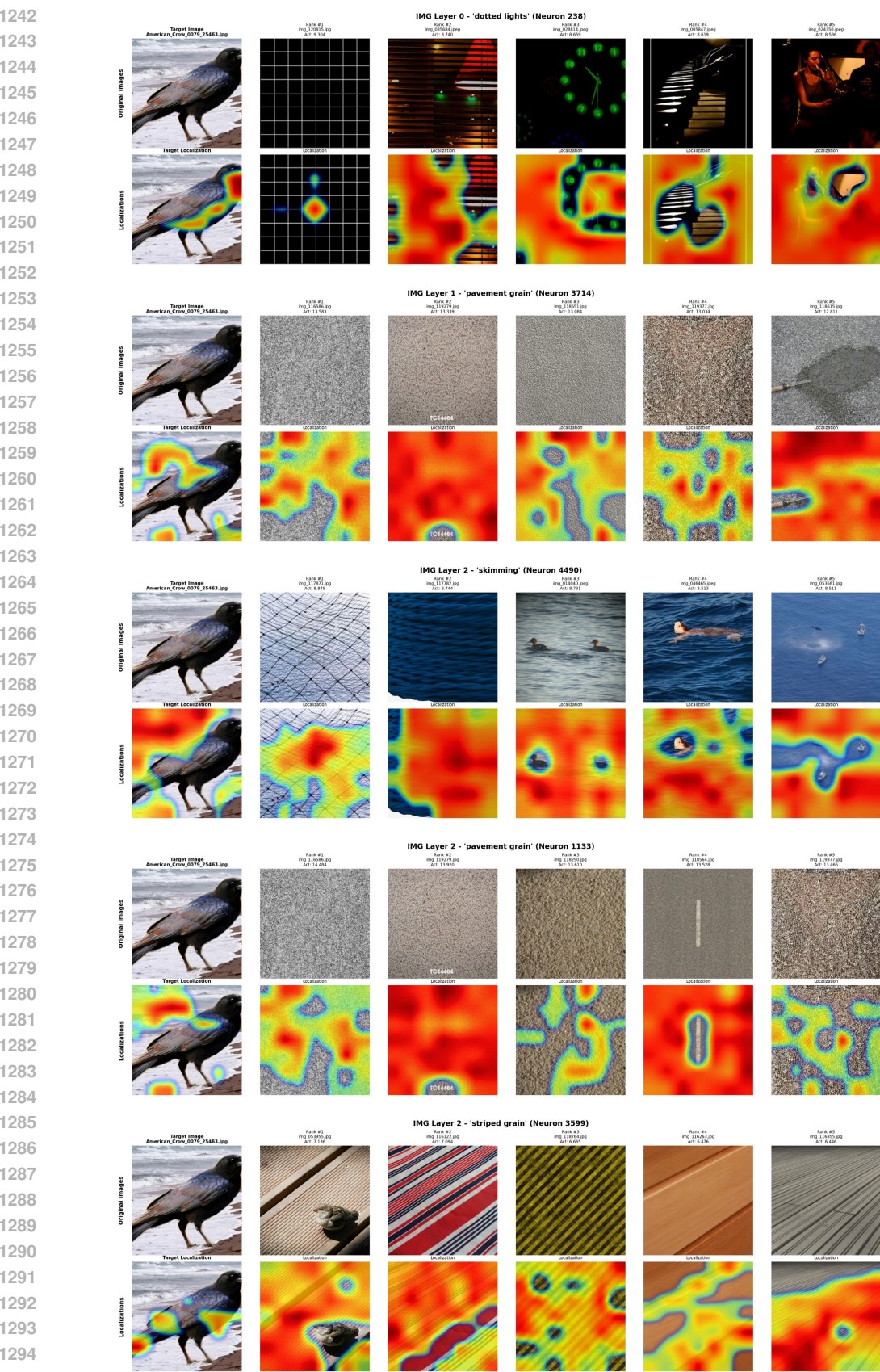

Figure 12: Concept localizations for landbird on water background.

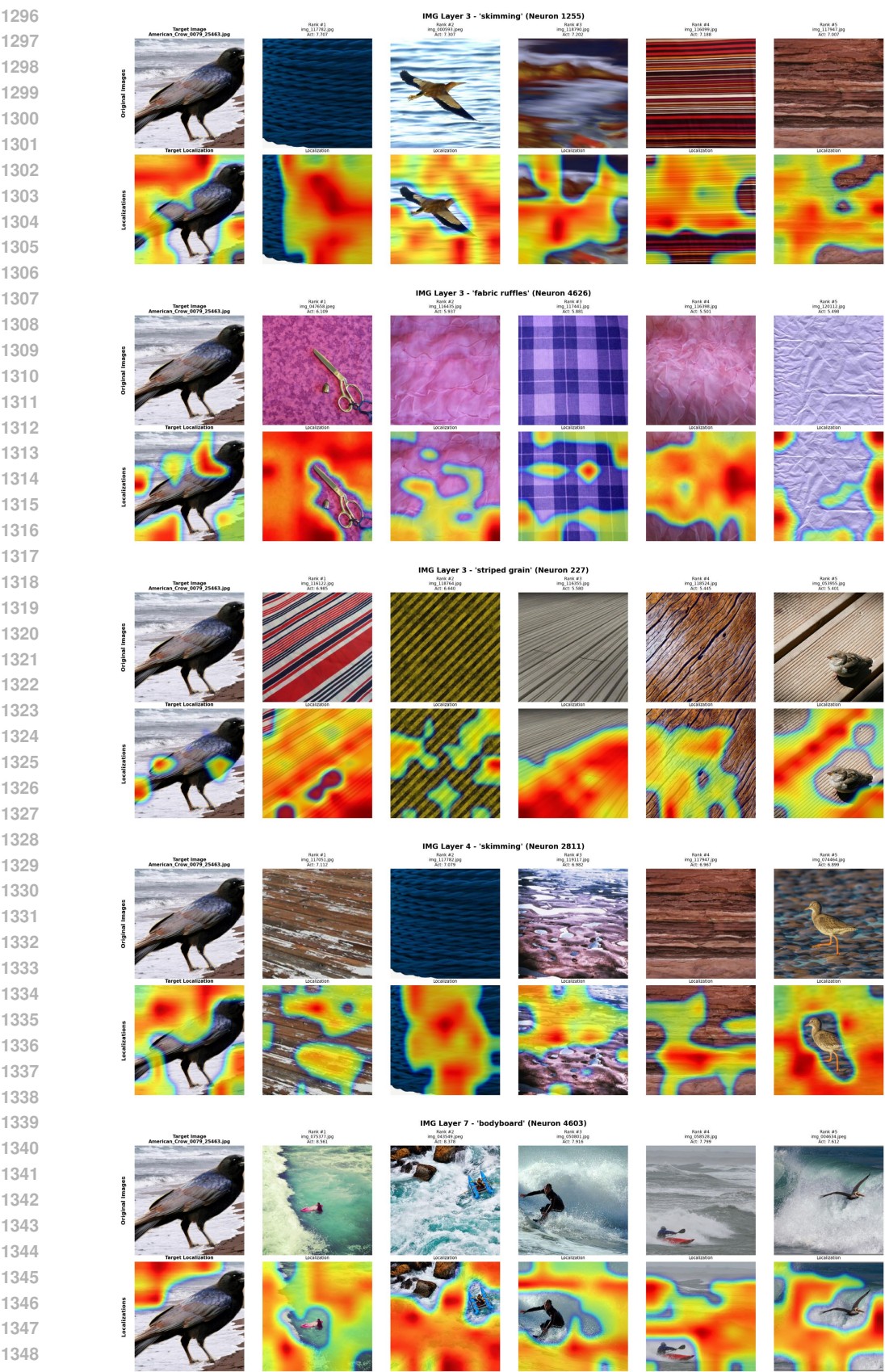

Figure 13: Concept localizations for landbird on water background.

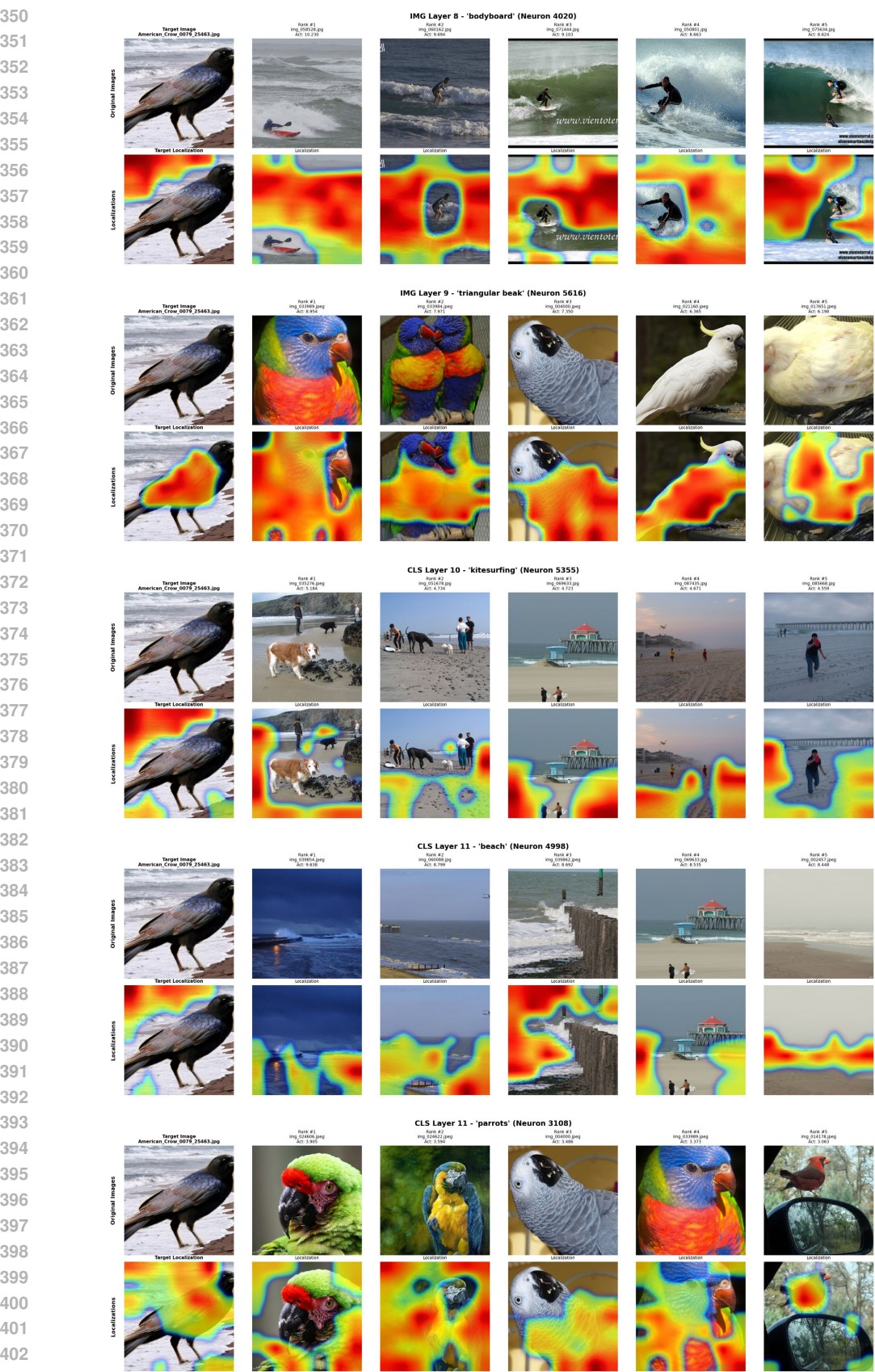

Figure 14: Concept localizations for landbird on water background.

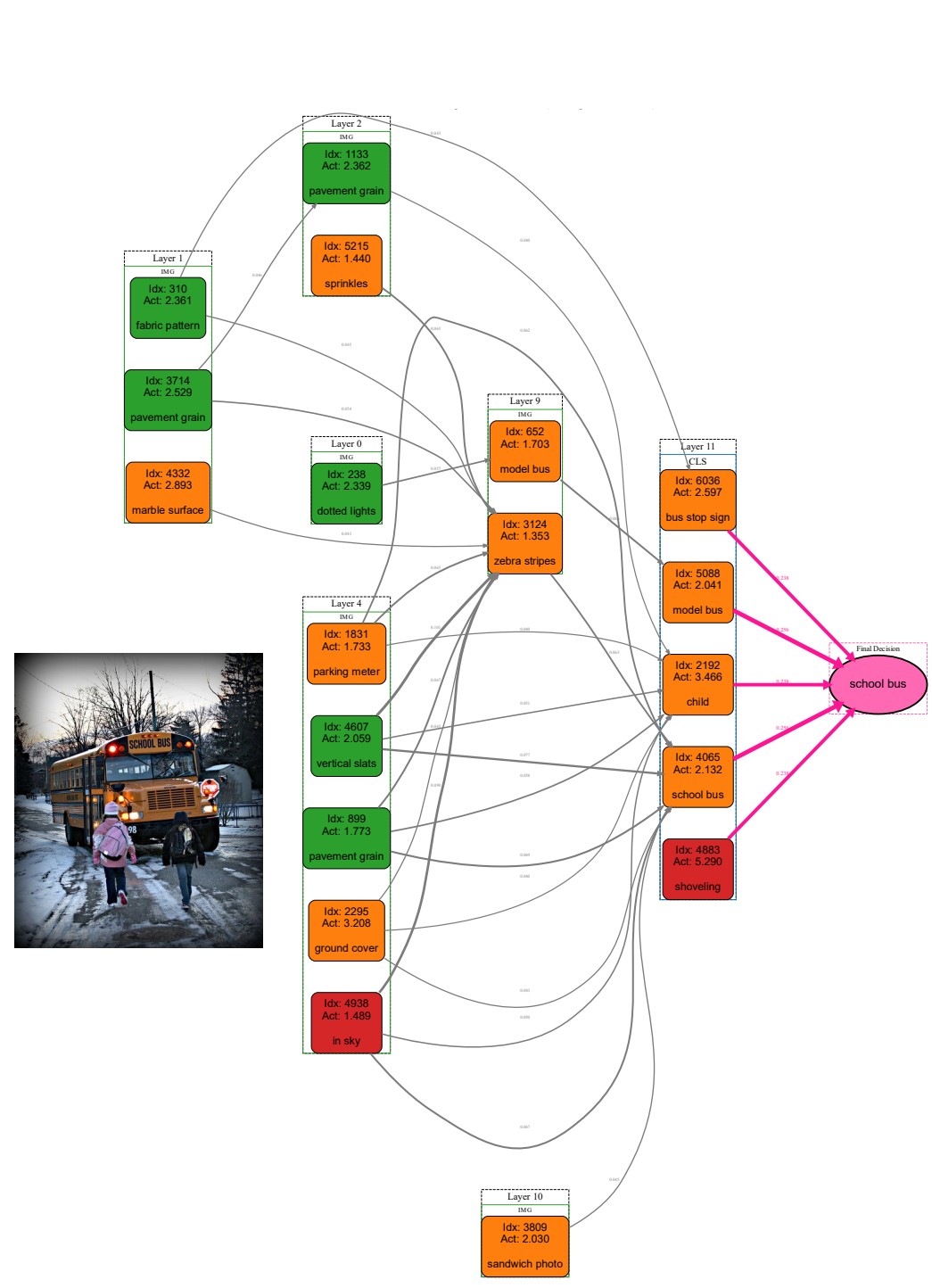

Figure 15: Concept circuit for school bus.

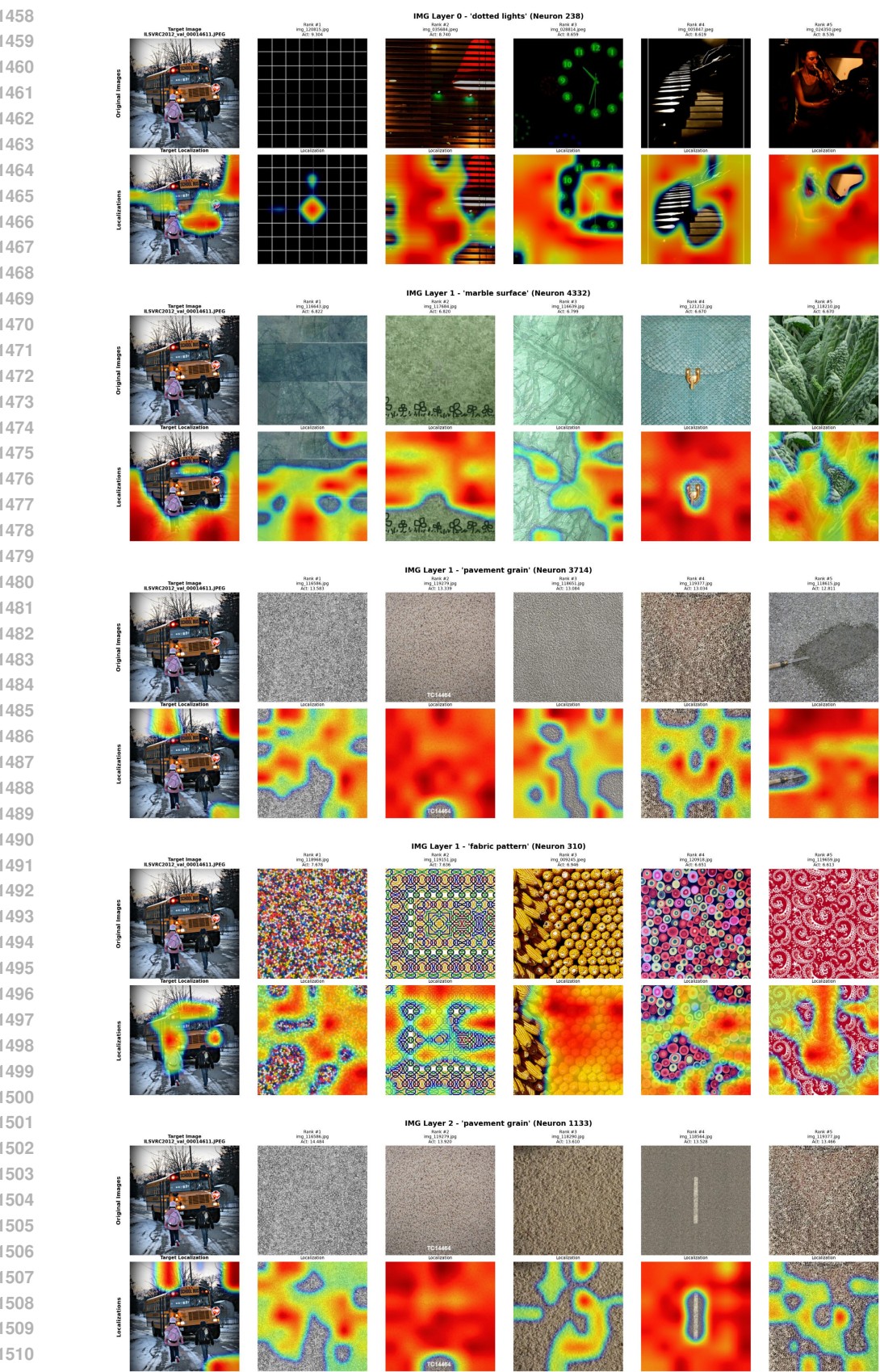

Figure 16: Concept localizations for school bus.

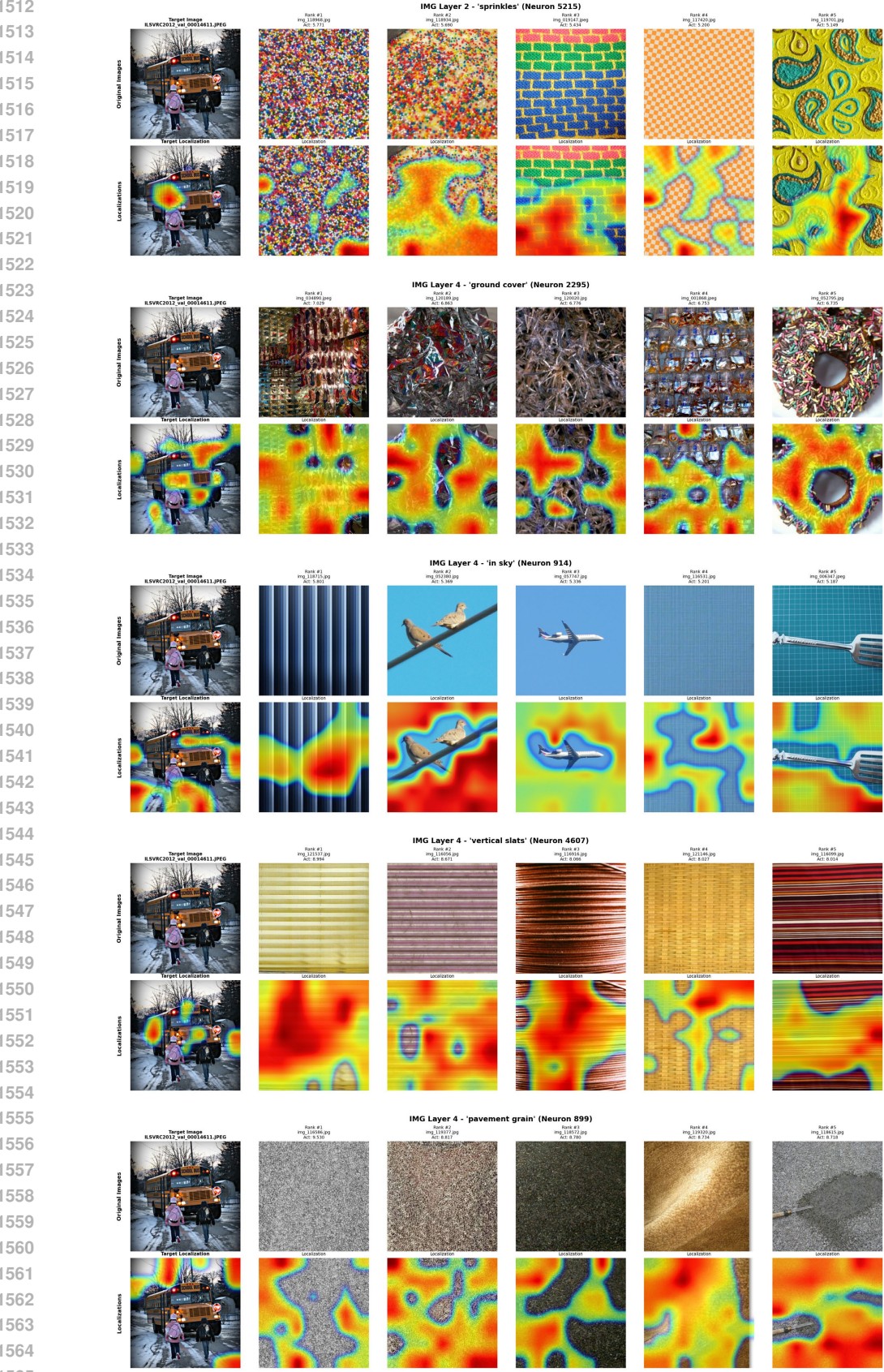

Figure 17: Concept localizations for school bus.

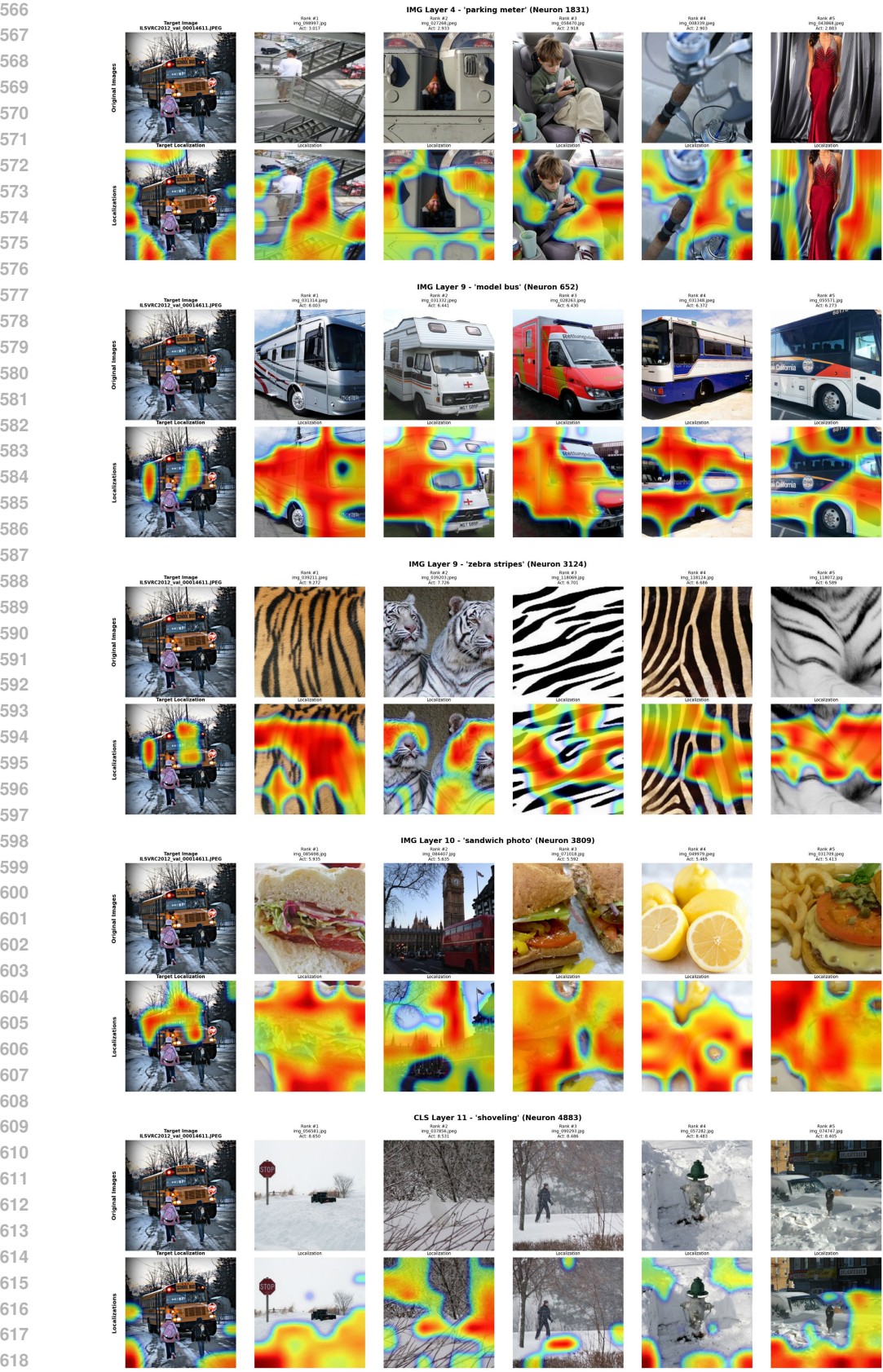

Figure 18: Concept localizations for school bus.

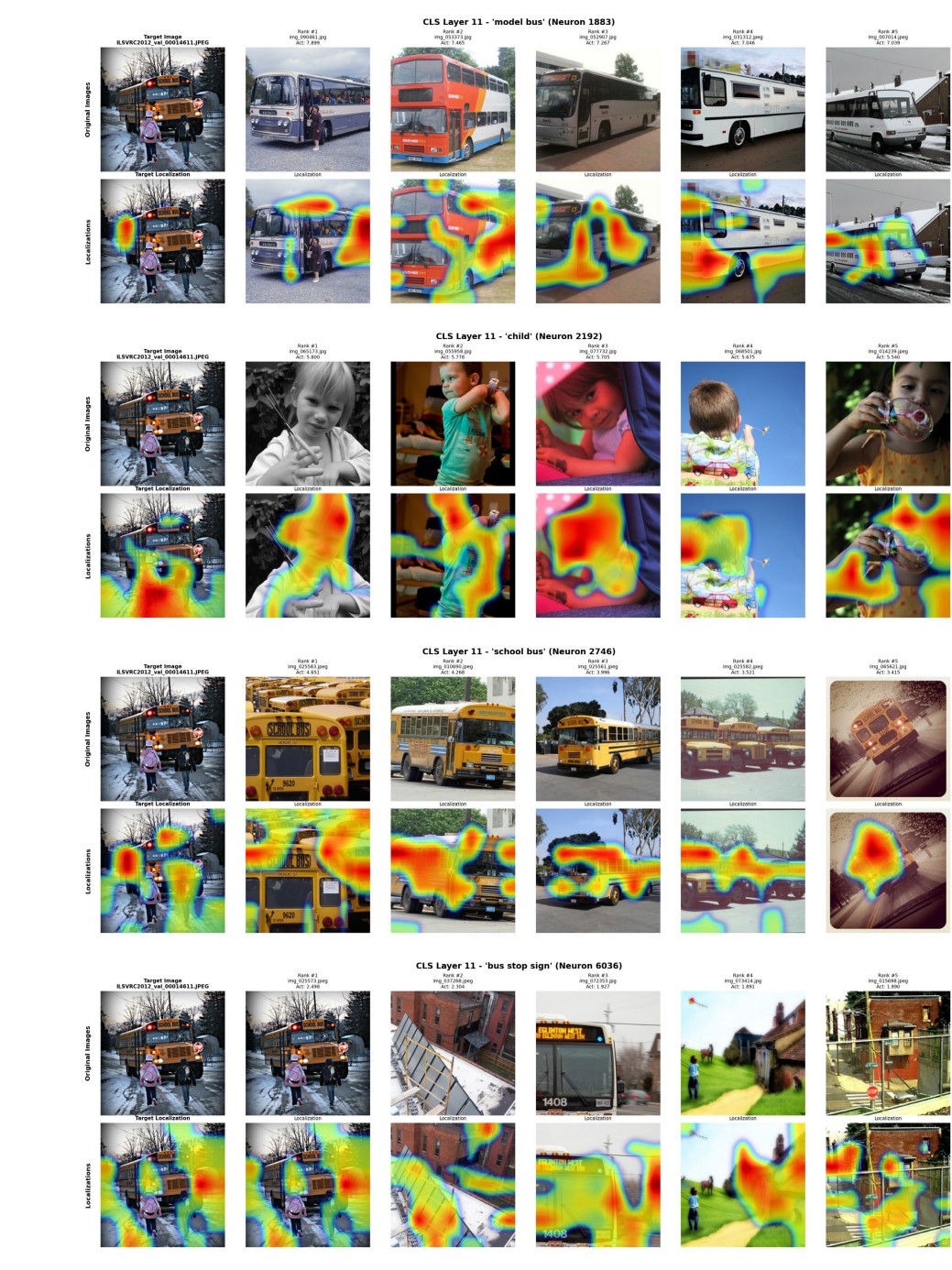

Figure 19: Concept localizations for school bus.

