# OpenReview forum: "Tracing Concept Circuits to Audit and Steer Vision Transformers"
_ICLR.cc/2026/Conference — Submitted to ICLR 2026_

### Official Review · Reviewer_D6zH · 2025-10-24

**Soundness:** 2
**Presentation:** 2
**Contribution:** 1
**Rating:** 2
**Confidence:** 4

**Summary:**

This work introduces a intepretability toolbox entitled ViSAE. The toolbox relies on Sparse Autoencoders (SAEs) to create monesemantic concepts that can later be combined with circuit discovery methods to trace decision throughout the network. The contributions of the work are along three axes, data, algorithm, and application. For the data, a new probing dataset is introduced using GPT-5 with the motivation of having more fine-grained concepts in the probing dataset compared to datasets such as Imagenet and MS-COCO. On the algorithm side, vision language models are used to label the concepts in the SAE, and a causal algorithm is used to trace concepts to the input. On the application side, the toolbox is demonstrated for the task of auditing and steering Vision Transfomers.

**Strengths:**

1. An extensive benchmarking of different types of SAE for vision data.
2. Nice figures both giving an overview of the work and showcasing the toolbox.
3. A clear list of contributions.

**Weaknesses:**

1. The novelty of the contributions is unclear.

(a) Data: A key contribution is the new dataset that provides more fine-grained concepts compared to the more object oriented datasets like ImageNet and MS-COCO. However, the BRODEN dataset [1] that is mentioned already provides more fine-grained concepts specifically designed for this type of analysis. It is unclear why this dataset is not used as a baseline or discussed further. Furthermore, using language models to generate fine-grained concepts is also an established practice, see for example [2].

(b) Algorithm: The contributions on the algorithm side are connected to the top-down concept reading and the bottom-up causal tracing. The top-down concept reading automatically labels the features of the SAE into labeled concepts, and the bottom-up causal tracing visualizes the concepts in the input space. But both of these algorithms appear to be direct applications of prior works. The top-down concept reading appear to be the CLIP-Dissect procedure [3]. The bottom-up causal tracing also seems to closely follow prior works [4, 5]. It is unclear what the methodological contributions of this work are on the algorithm side.

(c) Application: Both auditing and steering are tasks that are possible to do prior to the introduction of ViSAE [6, 7, 8].

2.The experimental evaluation is limited. Both evaluation methods and baselines are throughout the paper not suitable to demonstrate the potential quality of the introduced toolbox. The probing dataset used as baselines should be replaced with existing fine-grained concept-level probing datasets. The baselines in Table 4 could also be improved. CBM are old, and newer alternatives like Post-hoc CMBs [9] seem more relevant. SpLiCE is a good baseline, but not enough on its own, and comparing to other methods like [6, 7, 8] would strengthen the analysis. Furthermore, the visualization in Figure 5 and 6 are nice, but are only qualitative. Established quantitative measures [10] should be used to evaluated the visualizations.

- [1] Bau et al., Network Dissection: Quantifying Interpretability of Deep Visual Representations, CVPR 2017
- [2] Yang et al., Language in a Bottle: Language Model Guided Concept Bottlenecks for Interpretable Image Classification, CVPR 2023
- [3] Oikarinen et al., CLIP-Dissect: Automatic Description of Neuron Representations in Deep Vision Networks, ICLR 2023.
- [4] Commy et al., Towards Automated Circuit Discovery for Mechanistic Interpretabiliy, NeurIPS 2023
- [5] Meng et al., Locating and Editing Factual Associations in GPT, NeurIPS 2022
- [6] Dreyer et al., Mechanistic understanding and validation of large AI models with SemanticLens, Nature Machine Intelligence 2025
- [7] Wu et al., Discover and Cure: Concept-aware Mitigation of Spurious Correlation, ICML 2023
- [8] Joseph et al, Steering CLIP's vision transformer with sparse autoencoders, CVPR Workshop on Mechanistic Interpretability for Vision 2025
- [9] Yuksekgonul et al., Post-hoc Concept Bottleneck Models, ICLR 2023
- [10] Hedström et al., Quantus: An Explainable AI Toolkit for Responsible Evaluation of Neural Network Explanations and Beyond, JMLR 2023

**Questions:**

1. How does the proposed new dataset compare to [1] and [2]?
2. How does the results look if [1] and [2] are used Table 3 and 5?
3. What is the difference between the CLIP-Dissect procedure and top-down concept reading?
4. What is the methodological novelty in the the bottom-up causal tracing?
5. How does the auditing and steering results look compared to other more recent and relevant baselines?
6. The visualizations in Figure 6 look good at first glance, but when looking closer they seem inconsistent. In the "lines" example, the woman's necklace is highlighted, but the straight black line of her dress is somehow not a "line". The "wooden" example has parts of a wooden fence highlighted, why not the rest? Similar inconsistencies appear in the other examples. How can we understand these inconsistencies?

- [1] Bau et al., Network Dissection: Quantifying Interpretability of Deep Visual Representations, CVPR 2017
- [2] Yang et al., Language in a Bottle: Language Model Guided Concept Bottlenecks for Interpretable Image Classification, CVPR 2023

---

> ### Author Response · Authors · 2025-11-25
> **Rebuttal by Authors (Part 1/2)**
>
> Thank you for your detailed and instructive reviews. Below, we provide point-to-point responses.
>
> **Q1. The novelty of the contributions is unclear.**
> Our novelty is not another SAE variant. Instead, we introduce **a toolbox for interpreting the inner workings of ViTs:** a faithful **infrastructure** that is largely **missing** in the existing literature. The toolbox consists of: **(i) a neuroscience-motivated probing suite** of 64K images and 16K visual concepts, **(ii)** a **top-down** concept reading algorithm, and **(iii)** a **bottom-up** concept circuit tracing algorithm. Here, SAEs serve as one component within the concept-reading stage, rather than the main contribution. Building such a toolbox is non-trivial, and we address two concrete gaps:
>
> **(1) What probing data enables faithful interpretation of ViT inner workings?** Recent studies [1] show that what SAEs can see and explain is shaped by the data they are trained on and interpret. However, existing vision SAE works [2,3,4] often rely on off-the-shelf image sets (e.g., ImageNet) and generic text vocabularies. Consequently, their interpretations are typically skewed toward object-level concepts (Lines 65-70). To address this, motivated by the **human visual cortex hierarchy** from neuroscience [5], we construct a **first-of-its-kind probing suite** (64K images + 16K concepts) that ensures SAEs learn concepts spanning the full spectrum of visual processing. It covers concepts from low-level primitives (e.g., colors and edges), through intermediate textures and shapes, to high-level objects and parts, and finally scene-level motions and relations. Empirically, our probing images achieve **20× more efficient** concept coverage (table below), and our concept set enables **auto-interpretation** and improves its accuracy by 28.7% over existing vocabularies (Tab. 3, rows 3-5).
>
> | Probing Image Set | # of Images | Concepts Covered by Images (%) |  |  |  |  | Coverage Efficiency $\uparrow$ (%/1K Images) |
> |---|---:|---:|---:|---:|---:|---:|---:|
> |  |  | Primitive | Intermediate | Object | Scene | Avg. |  |
> | ImageNet | 1,281K | 81.0 | 78.2 | 97.7 | 59.0 | 78.9 | 0.06 |
> | MSCOCO | 118K | 69.6 | 65.4 | 80.4 | 63.1 | 69.6 | 0.59 |
> | Ours | 64K | 87.1 | 80.6 | 92.6 | 61.7 | **80.5** | **1.26** |
>
> **(2) How to represent the inner workings of ViT in a human-readable way?** Existing vision SAE works typically extract **discrete concepts from a single layer,** often the final embeddings (Lines 70–74). As a result, they miss a key aspect of ViT inner workings: how concepts interact and propagate across layers. To fill this gap, our toolbox provides (i) a top-down step that reads human concepts from ViT representations, and (ii) a bottom-up step that traces their cross-layer interactions. Together, our toolbox offers **structured cross-layer concept circuits** that capture how concepts emerge, evolve, and interact to jointly drive predictions. This mechanism view significantly benefits downstream steering, improving worst-group accuracy by 33.7% (Tab. 4).
>
> &nbsp;
>
> **Q2. The differences between CLIP-Dissect and our top-down concept reading.**
> We adapt CLIP-Dissect to automatically interpret SAE features and differ from the original CLIP-Dissect in two aspects. (1) CLIP-Dissect directly labels model neurons without resolving the **polysemantic neuron** issue (superposition). We instead adapt their scoring procedure in the SAE feature space, where sparse coding yields approximately** monosemantic features.** (2) CLIP-Dissect relies on the Google-20K vocabulary, which is biased toward **linguistically frequent words** and loosely tied to image content (Lines 190–194). In contrast, our **visual-grounded concept** set is constructed from images and leads to substantially higher interpretation accuracy.

---

> ### Author Response · Authors · 2025-11-25
> **Rebuttal by Authors (Part 2/2)**
>
> **Q3. Why haven't all the concept-related areas been highlighted in Fig.6?**
> This is an interesting observation. We provide two complementary perspectives.
> **(1) Resolution of the explainee ViT.** Our explainee backbone is ViT-B/32, which splits the image into a coarse 7×7 grid of tokens. Each token, therefore, covers a relatively large region that often contains multiple concepts. In the “wooden fence” example you mentioned, the patches that are not highlighted typically include a mix of “wooden” plus other concepts. For those tokens, the activation values of the “wooden” concept fall below the visualization threshold. **(2) Granularity of concepts and embeddings** (Line 480-485). The automatic interpretation is also constrained by the granularity of both the concept set and the VLM embedding space (CLIP-Dissect). An SAE feature labeled as “wooden” is not necessarily a pure “all wooden pixels” detector. It may correspond to a finer-grained concept, such as “vertical wooden slats with shadow” or “weathered brown wood.” In such cases, only the regions that match this finer-grained pattern are strongly activated, but it will still be named as “wooden” because of the interpretation granularity.
>
> &nbsp;
>
> **Q4. Additional evaluation and baselines.**
> 1. **Auditing comparison with Network Dissection (ND) [6] and Language in a Bottle (LaBo) [7].** For ND, we train SAEs on their images and interpret using their concept set; For LaBo, we use their concept set to interpret our trained SAEs. As shown in the table below, both our probing images and concepts outperform ND and LaBo. Note that LaBo does not provide a new probing image set, so we use ImageNet and LaBo’s concept set for ImageNet.
>
> |                   |               | Interpretation Accuracy (%) |          |          |
> |-------------------|---------------|--------------------------|----------|----------|
> | Probing Image Set |  Concept Set  |          Top-10          |  Top-20  |  Top-30  |
> |       Broden      |     Broden    |           16.8           |   23.9   |   26.8   |
> |      ImageNet     | LaBo-ImageNet |           15.9           |   18.4   |   20.3   |
> |        Ours       |      Ours     |         **36.6**         | **47.6** | **54.1** |
>
> 2. **Steering comparison with Posthoc CBM (PCBM) [8] and DN-CBM [9].** As shown in the table below, our method significantly outperforms PCBM and DN-CBM on the steering results for the Waterbird dataset.
>
> | Method | CBM | SpLiCE | DN-CBM | PCBM | Ours |
> |---|---|---|---|---|---|
> | Worst Group Acc. (before steer) | 37.3 | 48.0 | 57.5 | 50.3 | 50.3 |
> | Worst Group Acc. (after steer) | 51.8 | 60.0 | 71.3 | 74.7 | **98.5** |
> | $\Delta$ | +14.5 | +12.0 | +13.8 | +24.4 | **+48.2** |
>
> 3. **Quantitative measures of concept localization using Quantus [10].** As shown in the table below, our concept localization outperforms the baseline attribution method [11] on localization accuracy by 3.7% using the VOC2007 dataset.
>
> | Method        | Point Game | Attribution Localization |
> |---------------|:----------:|:------------------------:|
> | Chefer et al. |    41.9    |           32.3           |
> | Ours          |  **45.0**  |         **36.0**         |
>
> &nbsp;
>
> *We have incorporated all of these changes into the revised manuscript (highlighted in blue). We believe these clarifications and additional results address your concerns. We kindly ask the reviewer to take them into account when considering any score updates. We welcome any further questions or feedback.*
>
> &nbsp;
>
> [1] Hindupur et al. "Projecting assumptions: The duality between sparse autoencoders and concept geometry." ICML’25W.
> [2] Lim et al. "Sparse autoencoders reveal selective remapping of visual concepts during adaptation." ICLR’25.
> [3] Thasarathan et al. "Universal sparse autoencoders: Interpretable cross-model concept alignment." ICML’25.
> [4] Rao et al. "Discover-then-name: Task-agnostic concept bottlenecks via automated concept discovery." ECCV’24.
> [5] Goodale et al. "Separate visual pathways for perception and action." Trends in Neurosciences, 1992.
> [6] Bau et al., Network Dissection: Quantifying Interpretability of Deep Visual Representations, CVPR’17.
> [7] Yang et al., Language in a Bottle: Language Model Guided Concept Bottlenecks for Interpretable Image Classification, CVPR’23.
> [8] Yuksekgonul et al., Post-hoc Concept Bottleneck Models, ICLR’23.
> [9] Rao et al. "Discover-then-name: Task-agnostic concept bottlenecks via automated concept discovery." ECCV’24.
> [10] Hedström et al., Quantus: An Explainable AI Toolkit for Responsible Evaluation of Neural Network Explanations and Beyond, JMLR’23.
> [11] Chefer et al. "Generic attention-model explainability for interpreting bi-modal and encoder-decoder transformers." ICCV’21.

---

### Official Review · Reviewer_DS6U · 2025-10-27

**Soundness:** 3
**Presentation:** 4
**Contribution:** 3
**Rating:** 6
**Confidence:** 4

**Summary:**

This paper focuses on the interpretability of Vision Transformers (ViTs) and contributes with the following three angles. First, it constructs a dataset from 7 sources and annotates the images with concepts inspired by neuroscience. Second, based on the current literature of Sparse Autoencoders (SAEs), it proposes a reading algorithm to assign an SAE feature a specific concept label based on the (image, concept) probing set, and uses an existing tracing algorithm to build connection graphs between SAE concepts. Third, it conducts experiments to interpret the information flow during ViT decision making and to steer model behavior by editing concepts through SAE interventions.

**Strengths:**

The paper is well written and is easy to read.

The fine-grained annotation significantly helps ViT interpretability through SAEs.

With the help of concept annotation, the top-down reading algorithm avoids human labor or imprecise summaries from models for SAE feature label assignment. Further, broader concepts allow for detailed diagnosis of failure modes and bring practical benefits, as reflected in the impressive steering outcomes. Such fine-grained concepts allow for discovering new connection graphs, which brings clear future research benefits.

**Weaknesses:**

The main weakness of the paper I spot is the extent of technical contribution. It seems the improvement mainly comes from the GPT-5 annotation process, which is technically incremental. For example, as shown in both Table 3 and Table 5, the dataset's quality is not that important - with the correct concepts being considered, using MSCOCO alone already achieves comparable results to the carefully curated dataset.

**Questions:**

(1) How are the numbers in Table 1 computed? What is the definition of “Concepts Covered by Images”?

(2) How do you obtain $c_m$ for computing $P_{nm}$? Is it obtained directly through the text description?

(3) In the experiment section for auditing, during the localization of concepts on pixels, how do you pick the layer to perform such attribution? How does this choice affect the heatmap? An ablation on this would be helpful.

(4) How do you launch the experiments in Section 3.2, part (2)? It seems there is a concept set mismatch for the ablation runs because the ground-truth concepts are obtained according to Section 2.2 (from the Ours-16K set) while the SAE concepts are obtained through ablation settings. If this is the case, the numbers reported are not meaningful, and thus the comparison is not valid.

(5) In Section 3.4, the accuracy reported in text (49.6) does not match that in the table (50.3). Why are they different? Also, similar to question (3), which layer do you use to pick the concept for steering? I suppose there are multiple SAE features that have been labeled as exactly the same background concept (because the total number of SAE features is likely exceeding the total number of available concepts). If this is the case, is intervention always effective regardless of the layer index?

(6) For section 3.2/3/4, are your SAE trained on cls token? Does switching to img token make a difference?

Misc: Table 1 seems to be missing SN under data source

---

> ### Author Response · Authors · 2025-11-25
> **Rebuttal by Authors (Part 1/2)**
>
> Thank you for your encouraging comments. Below, we provide point-to-point responses.
>
> **Q1. Is our work technically incremental?**
> We respectfully disagree. The reviewers’ novelty concerns mainly target SAE architecture or optimization, whereas our contribution is different in nature. Our novelty is not another SAE variant. Instead, we introduce **a toolbox for interpreting the inner workings of ViTs:** a faithful **infrastructure** that is largely **missing** in the existing literature. The toolbox consists of: **(i) a neuroscience-motivated probing suite** of 64K images and 16K visual concepts, **(ii)** a **top-down** concept reading algorithm, and **(iii)** a **bottom-up** concept circuit tracing algorithm. Here, SAEs serve as one component within the concept-reading stage, rather than the main contribution. Building such a toolbox is non-trivial, and we address two concrete gaps:
>
> **(1) What probing data enables faithful interpretation of ViT inner workings?** Recent studies [1] show that what SAEs can see and explain is shaped by the data they are trained on and interpret. However, existing vision SAE works [2,3,4] often rely on off-the-shelf image sets (e.g., ImageNet) and generic text vocabularies. Consequently, their interpretations are typically skewed toward object-level concepts (Lines 65-70). To address this, motivated by the **human visual cortex hierarchy** from neuroscience [5], we construct a **first-of-its-kind probing suite** (64K images + 16K concepts) that ensures SAEs learn concepts spanning the full spectrum of visual processing. It covers concepts from low-level primitives (e.g., colors and edges), through intermediate textures and shapes, to high-level objects and parts, and finally scene-level motions and relations. Empirically, our probing images achieve **20× more efficient** concept coverage (table below), and our concept set enables **auto-interpretation** and improves its accuracy by 28.7% over existing vocabularies (Tab. 3, rows 3-5).
>
> | Probing Image Set | # of Images | Concepts Covered by Images (%) |  |  |  |  | Coverage Efficiency $\uparrow$ (%/1K Images) |
> |---|---:|---:|---:|---:|---:|---:|---:|
> |  |  | Primitive | Intermediate | Object | Scene | Avg. |  |
> | ImageNet | 1,281K | 81.0 | 78.2 | 97.7 | 59.0 | 78.9 | 0.06 |
> | MSCOCO | 118K | 69.6 | 65.4 | 80.4 | 63.1 | 69.6 | 0.59 |
> | Ours | 64K | 87.1 | 80.6 | 92.6 | 61.7 | **80.5** | **1.26** |
>
> **(2) How to represent the inner workings of ViT in a human-readable way?** Existing vision SAE works typically extract **discrete concepts from a single layer,** often the final embeddings (Lines 70–74). As a result, they miss a key aspect of ViT inner workings: how concepts interact and propagate across layers. To fill this gap, our toolbox provides (i) a top-down step that reads human concepts from ViT representations, and (ii) a bottom-up step that traces their cross-layer interactions. Together, our toolbox offers **structured cross-layer concept circuits** that capture how concepts emerge, evolve, and interact to jointly drive predictions. This mechanism view significantly benefits downstream steering, improving worst-group accuracy by 33.7% (Tab. 4).
>
> &nbsp;
>
> **Q2. Why does MSCOCO have comparable results in Tabs. 3&5?**
> (1) The gains of the MSCOCO-trained SAEs in Tabs. 3&5 largely come from interpreting them with our concept set. When we instead interpret the same MSCOCO-trained SAEs using existing vocabularies (e.g., Google-20K / LAION-15K), performance drops substantially (see table below). (2) As shown in the table above, our probing images are twice as efficient in terms of concept coverage efficiency, which aligns with the Interpretation Accuracy results below.
>
> | Probing Image Set | Concept Set | Top-10 Interpretation Accuracy (IA) | Top-20 IA | Top-30 IA |
> |:---:|:---:|:---:|:---:|:---:|
> | MSCOCO | Google-20K | 10.0 | 13.3 | 15.3 |
> | MSCOCO | LAION-15K | 17.4 | 22.4 | 25.5 |
> | MSCOCO | Ours-16K | 34.9 | 45.0 | 51.2 |
> | Ours | Ours-16K | **36.6** | **47.6** | **54.1** |
>
> &nbsp;
>
> **Q3. In Tab. 1, what does “Concepts Covered by Images (%)” mean, and how are the numbers computed?**
> Definition (Lines 199-202): For each abstraction level (primitive, intermediate, object, scene), “Concepts Covered by Images (%)” denotes the percentage of concepts in our concept set that are covered in a dataset. Concretely, we (1) compute CLIP similarities between every image in the image set and every concept in our concept set; (2) regard a concept as covered by an image if their CLIP similarity is ≥ 0.28  (an empirically chosen threshold indicating high likelihood of presence); and (3) report, for each level, the fraction of concepts that are covered in this way.

---

> ### Author Response · Authors · 2025-11-25
> **Rebuttal by Authors (Part 2/2)**
>
> **Q4. Implementation details.**
> 1. **How do you obtain $c_m$ for computing $P_nm$? Is it from the text description?** Yes. Each concept $c_m$ is a one- or two-gram textual description (e.g., “striped fur”, “wooden texture”) in the concept set been used for interpretation (Ours-16K, LAION-15K, or Google-20K).
> 2. **Are the concept sets mismatched in Sec. 3.2 (part 2)?** No. Our metric is designed specifically to avoid such circularity. We use **semantic matching (CLIP-based), not string matching.** We consider a ground-truth concept “covered” if any concept in that vocabulary is semantically close in CLIP space, not necessarily if the string is identical.
> 3. **Layer choice for localization and its effect on heatmaps.** We do **not manually choose** the layer; the **SAE activations determine it.** For example, if an image strongly activates an SAE feature labeled “wooden texture” in its layer-3 representation, we use that layer-3 feature for localization. Specifically, we read out its activations on all layer-3 image tokens and convert them into a spatial heatmap (Lines 254–257). This indicates that the backbone ViT processes “wooden texture” information primarily at layer 3.
> 4. **Which layer do you use to pick the concept for steering?** We do **not manually select** a specific layer. Instead, with the help of our large-scale auto-interpretation, we can identify the SAE feature indices corresponding to background concepts (e.g., “grass”, “land”) **at every layer.** For steering, we reconstruct each layer’s representations (CLS and image tokens) using the corresponding SAEs while setting the activations of those background features to zero. We then forward these reconstructed representations layer-by-layer through the ViT. As a result, the final-layer CLS token is (approximately) free of background information and is used for classification.
> 5.** Are SAEs trained on CLS tokens only?** No. Our SAEs are trained on both CLS and image tokens separately for every ViT layer (Lines 294-297). Concretely, for a ViT-B/32 model with 12 layers, we train two SAEs (CLS and Image separately) per layer, and 24 SAEs in total.
>
> &nbsp;
>
> **Q5. Minors.**
> Thank you for catching the typos. We will add “SN” to the “Data Source” column of Tab.1. The numbers in the tables (50.3) are correct, we will update them in the main text.
>
> &nbsp;
>
> *We have incorporated all of these changes into the revised manuscript (highlighted in blue). We believe these clarifications and additional results address your concerns. We kindly ask the reviewer to take them into account when considering any score updates. We welcome any further questions or feedback.*
>
> &nbsp;
>
> [1] Hindupur et al. "Projecting assumptions: The duality between sparse autoencoders and concept geometry." ICML’25W.
> [2] Lim et al. "Sparse autoencoders reveal selective remapping of visual concepts during adaptation." ICLR’25.
> [3] Thasarathan et al. "Universal sparse autoencoders: Interpretable cross-model concept alignment." ICML’25.
> [4] Rao et al. "Discover-then-name: Task-agnostic concept bottlenecks via automated concept discovery." ECCV’24.
> [5] Goodale et al. "Separate visual pathways for perception and action." Trends in Neurosciences, 1992.

---

### Official Review · Reviewer_q9sp · 2025-10-30

**Soundness:** 3
**Presentation:** 3
**Contribution:** 2
**Rating:** 6
**Confidence:** 2

**Summary:**

The paper introduces concept circuits for ViTs: a layer-wise, directed graph where each node is a human-interpretable concept at a specific layer and edges indicate causal influence between concepts as they compose the final prediction. The authors 1) train sparse auto encoders per transformer block, 2) auto label each SAE feature with a text concept using a CLIP-based soft-WPMI alignment over a large, synthetic concept set, 3) perform within-model hard interventions by zeroing a source concept’s SAE activation and decoding, measuring the indirect effect on target concepts at later layers, and 4) create a layer-respecting DAG whose edge weights reflect these measured effects. Empirically, they show faithfulness via targeted ablations and qualitative circuits.

**Strengths:**

- Tackles a core gap in ViT interpretability: concept-level, layer-wise circuits.
- Uses within-model interventions rather than correlational probes.
- Clean, modular pipeline: per-layer SAEs on the residual stream + automated concept labeling.
- Produces DAG that attempts to capture compositional flow, providing a computation path that's interpretable.
- Enables steering by suppressing spurious concepts or amplifying robust ones.
- Targeted ablations yield consistent logit drops aligned with the inferred circuits.
- Overall, a good addition to the interpretability arsenal.

**Weaknesses:**

- The 16K concept set is produced by GPT-5, but there’s no systematic human audit.
- Prompts are given, but generation settings (e.g., temperature/seed/determinism) aren’t specified, so exact reproduction of the concept set isn’t guaranteed. As far as I know, GPT5’s API does not provide a deterministic option at all.
- Eq 4 clearly implies that this mapping is neither injective nor surjective. This is only briefly acknowledged and not sufficiently analyzed.
- The do operator is used to define indirect effects via within-model edits, but no explicit SCM/graph is specified.
- Table 3 shows that “Ours-16k” greatly improves accuracy when the ground truth for the test split is itself GPT-5 concept annotations drawn from their proving suite. This probably privileges their vocabulary vs LAION/Google.
- The alignment step assumes CLIP’s embedding geometry reflects concept presence.

Minor: The X-ray/MRI analogy is confusing and potentially misleading.

**Questions:**

- What explicit SCM (variables, structural assignments) underlies Eq 5 and 6, and under which assumptions are the reported indirect effects identified? Do you treat the forward computation graph as the causal graph? Please specify.
- How are interventions kept on-manifold when zeroing SAE features and decoding?
- How often do multiple SAE features map to the same concept, and how many concepts receive none? What is the impact on edge weights and interpretation accuracy?
- What sampling settings produced the 16K concept list? Is the list reproducible?
- How is vocabulary circularity ruled out in 3.2/Table 3?
- To ensure acyclicity, edges must be $s\to t$ with $t>s$. Can you state this explicitly?
- Do steering edits transfer beyond WaterBirds?
- Did you try not pruning the initial pool?

---

> ### Author Response · Authors · 2025-11-25
> **Rebuttal by Authors**
>
> We appreciate your encouraging comments and recognition of our contributions. Below, we provide point-to-point responses.
>
> **Q1. How was the accuracy and reproducibility of GPT-5 annotations ensured?**
> We ensure the accuracy and reproducibility along three dimensions. (1) Careful prompt engineering (full prompt template in Appendix B) to ensure the LLM follows the instructions, generating consistent concept annotations. (2) Full details of the generation hyperparameters for the reproducibility, such as temperature=1.0, seed=42. (3) Additional human evaluation results of our concept set. As shown below, our concept set is faithful to the image and comprehensive across abstraction levels (>4.7/5 Likert score).
>
> | Evaluator    |     Human_A     |     Human_B     |     Human_C     | Avg. |
> |--------------|:---------------:|:---------------:|:---------------:|:----:|
> | Faithfulness | 4.65 $\pm$ 0.06 | 4.83 $\pm$ 0.20 | 4.90 $\pm$ 0.05 | 4.79 |
> | Completeness | 4.72 $\pm$ 0.04 | 4.73 $\pm$ 0.06 | 4.78 $\pm$ 0.18 | 4.74 |
>
> - Evaluation Data: Three groups of random images (3X50) paired with our concept annotations.
> - Metrics: Faithfulness (whether concepts are visually grounded in the image) and Comprehensiveness (whether they cover all visual abstraction levels).
> - Scoring: Both metrics are rated on a 0-5 Likert scale (0 = completely wrong, 5 = perfect match).
> - Evaluators: Three Computer Science major graduate students.
> - Conclusions: (1) The overall quality of our concepts is high. (2) Our concept annotation process is consistent across images.
>
> &nbsp;
>
> **Q2. What is the impact of non-injective and non-surjective mapping in Eq. 4?**
> This behavior is intentional rather than a flaw, and reflects two design choices. (1) Multiple SAE features are allowed to map to the same concept because different units can specialize in different variants or contexts of that concept (e.g., “stripe” on fur vs. “stripe” on clothing). (2) Our concept vocabulary is **intentionally over-complete.** It contains many more candidate concepts than any single model will actually use. This is expected and even desirable: the circuits should reflect which concepts the backbone explainee model actually encodes and relies on, not the entirety of the vocabulary.
>
> &nbsp;
>
> **Q3. Do you treat the forward computation graph as the causal graph?**
> Yes. We model the ViT forward pass as a deterministic Structural Causal Model (SCM), and our concept nodes just make it a **finer-grained** SCM built on top of it. Under this setup, the indirect effects in Eqs. 5&6 are identified under the following assumptions: (1) The network is feed-forward and acyclic, so the forward pass defines a DAG. (2) The model and SAE are deterministic functions of their inputs (no hidden noise or unobserved confounders between layers). (3) The SAE-decoded representation stays close to the original manifold (verified by low reconstruction error).
>
> &nbsp;
>
> **Q4. Implementation details.**
> 1. **How are interventions kept on-manifold?** Our interventions are applied in the SAE latent space and are designed to stay close to the SAE’s activation manifold. We use a BatchTopK SAE, so for each image, only the top-K features have non-zero activations, and all remaining features are zero during decoding. In other words, the decoder is explicitly trained to reconstruct representations from very sparse codes where most coordinates are zero. After zeroing part of the feature activations, the resulting code is still a valid sparse code of the same form seen during training. So the decoded representation remains on (or very close to) the model’s representation manifold.
> 2. **How is vocabulary circularity ruled out in 3.2/Table 3?** Our metric is designed specifically to avoid such circularity. We use semantic matching (CLIP-based), not string matching. We consider a ground-truth concept “covered” if any concept in that vocabulary is semantically close in CLIP space, not necessarily if the string is identical.
> 3. **Does steering transfer beyond WaterBirds?** Yes. As shown in the table below, we conduct the linear probe using CLIP's last layer representations. Our method can surgically target and remove information pertaining to glasses and reduce classifier performance while preserving information relevant to gender classification. Our method outperforms the baseline SpLiCE method by 12% accuracy in terms of steering power on this task.
>
> |  | Gender Acc. |  |  | Glasses Acc. |  |  |
> |---|---|---|---|---|---|---|
> |  | Before Steering | After | $\Delta$ $\downarrow$ | Before | After | $\Delta$ $\uparrow$ |
> | SpLiCE | 0.89 | 0.85 | 0.04 | 0.88 | 0.59 | 0.29 |
> | Ours | 0.98 | 0.96 | **0.02** | 0.96 | 0.55 | **0.41** |
>
> &nbsp;
>
> **Q5. Minors.**
> We appreciate the comments on our paper presentation, such as the X-ray/MRI analogy could be confusing, and the explicit statement of s->t, t>s. *We have applied these suggestions and the discussion above in the revised version (highlighted in blue).*

---

### Official Review · Reviewer_CZUB · 2025-10-31

**Soundness:** 1
**Presentation:** 2
**Contribution:** 2
**Rating:** 2
**Confidence:** 5

**Summary:**

This paper introduces ViSAE, an interpretation toolbox for auditing and steering ViTs by tracing human-understandable concept circuits within their internal representations. While the presented method addresses an important problem and has many interesting components, the novelty and technical rigor are limited.

**Strengths:**

- The paper addressed the important problem of AI safety by building a framework that integrates multiple components, such as data curation, SAE methods, and use cases.
- The concept dataset presented in the paper would be a great resource for future research on developing concept-based explanation methods.

**Weaknesses:**

- The technical novelty is limited. There is prior work published in ECCV 2024 (https://arxiv.org/abs/2407.14499) of which core technique largely overlaps with that of this paper. This prior work has not been cited or discussed.
- The probing image set is limited. I agree that ImageNet is too focused on object-level tasks, and an alternative dataset is required. However, the probing image set of 64K images, while carefully curated and outperforming existing smaller datasets, is still relatively small for training SAEs to capture the full spectrum of visual concepts, especially when compared to the vast datasets (e.g., LAION with billions of images) used for pre-training large vision models like CLIP. Usually, for LLMs, SAEs are trained on huge pre-training datasets.
- The authors measured monosemanticity by looking into "whether each basis feature of an SAE consistently activates on images of the same semantics" I am not sure if this metric is suited for measuring monosemanticity. “Monosemanticity” means that *each feature* in the sparse code represents *only one distinct concept*, even if multiple concepts are present in an image. The current metric might not fully verify if a single feature is truly disentangled from co-occurring concepts within an image, but rather if the *set of images* activating it are similar.
- The analogy is misleading. I appreciate the authors coming up with an analogy to explain how their work is different from prior work, which I believe is a good practice. However, I think the analogy is misleading in the context of the paper. X-rays and MRIs both provide spatial representations of physical structures, differing in dimensions. In contrast, pixel attribution methods provide spatial saliency maps on the input, whereas concept circuits provide causal graphs of abstract, or even non-localizable *concepts* within a model's latent space. The shift is not merely from "average" to "slice-by-slice" spatial views, but from input-level spatial attribution to a graph of abstract causal relationships, which fundamentally differs in its nature.
- The labeling of the two steps in the tracing algorithm as "Top-down concept reading" and "Bottom-up causal tracing" can be confusing. "Top-down" typically implies moving from higher-level to lower-level information, while "Bottom-up" means the reverse. In this context, "concept reading" involves mapping latent features (mid-level) to human concepts (high-level), which could be seen as an interpretation *of* features. "Causal tracing" explicitly builds connections from earlier (lower) layers to later (higher) layers and ultimately the prediction, which is indeed "bottom-up." Clarifying the precise meaning of "top-down" in "concept reading" or refining the terminology could improve clarity.

**Questions:**

- The process for visualization (e.g., pruning nodes) has not been mentioned. For example, in figure 5, how were the edges and nodes to show determined?
- GPT-5 was used for automatic concept annotation by GPT-5. How was the accuracy of the process ensured?

---

> ### Author Response · Authors · 2025-11-25
> **Rebuttal by Authors (Part 1/2)**
>
> Thank you for your detailed and instructive reviews. Below, we provide point-to-point responses.
>
> **Q1. Do our contributions overlap with ECCV’24?**
> We respectfully disagree. The referenced ECCV’24 paper is Discover-then-Name: Task-Agnostic Concept Bottlenecks via Automated Concept Discovery (DN-CBM) [1]. This work focuses on interpreting the final CLIP vision embeddings via a concept bottleneck model. We already discussed a series of SAE-based interpretation methods (Lines 70–72) similar to DN-CBM. In the revision, we will explicitly cite DN-CBM and expand this discussion. However, more importantly, the core technical contributions of our ViSAE **differ in both goal and methodology.** We have provided our major novelty in the general response, here are more detailed comparisons:
>
> 1. **Single-layer CBM vs. Cross-layer concept circuits.**
> DN-CBM trains a single SAE on the **final-layer CLIP embeddings** and uses its hidden units as a single concept bottleneck layer for classification. The object of its interpretation is the concept-bottleneck classifier, not the original ViT backbone. In contrast, we train multiple SAEs on the residual stream of **every ViT layer (both image and CLS tokens)** and trace directed, layer-by-layer concept circuits within the original ViT. This enables us to study how concepts emerge and transform across depth, and to use these circuits for auditing and steering the original foundation model (Sec. 2.3), which DN-CBM does not address.
> 2. **Final-embedding-only naming vs. Token-agnostic concept naming.**
> DN-CBM names concepts by matching SAE decoder dictionary vectors directly to CLIP text embeddings. However, this can work to interpret **only the CLIP final embeddings,** because all image tokens and the CLS tokens from other layers are not aligned in the vision-language embedding space. SAE features trained on these representations cannot be directly projected. In contrast, our method is **model-agnostic** and names SAE features **across layers and token types** (both image and CLS tokens). This design is crucial for interpreting and tracing concepts throughout the entire ViT, not only at the final embedding.
> 3. **Generic text-mined vocabularies vs. Neuroscience-motivated probing suite.**
> DN-CBM names concepts using a vocabulary of 20k frequent English words mined from text (i.e., the Google-20K concept set we already compared with). This vocabulary is not organized by visual abstraction and includes many **non-visual or weakly visual words** (e.g., “forte”, “constants”). Our concept set instead introduces ~16K visual concepts that are **grounded on our probing images,** which are explicitly organized into four neuroscience-motivated abstraction levels (primitive, intermediate, object, and scene). As shown in Table 3 (Sec. 3.2), when interpreting the same trained SAE, our concept set outperforms the 20K frequent English words vocabulary by 37.4% in terms of interpretation accuracy.
>
> - Additional comparison between steering power on the Waterbird dataset (DN-CBM vs Ours):
>
> | Method | CBM | SpLiCE | DN-CBM | PCBM | Ours |
> |---|---|---|---|---|---|
> | Worst Group Acc. (before steer) | 37.3 | 48.0 | 57.5 | 50.3 | 50.3 |
> | Worst Group Acc. (after steer) | 51.8 | 60.0 | 71.3 | 74.7 | **98.5** |
> | $\Delta$ | +14.5 | +12.0 | +13.8 | +24.4 | **+48.2** |
>
> &nbsp;
>
> **Q2. Is our probing image set of 64K limited compared to LAION / CC3M?**
> No. Our goal is **not to re-pretrain CLIP,** but to probe a fixed ViT with an image distribution that covers a wide spectrum of visual abstractions. In this case:
>
> 1. Our scale is large compared to existing fine-grained interpretability datasets. Network Dissection’s Broden dataset [2], widely used for feature interpretability, contains 63K images and 1,197 concepts. The Describable Textures Dataset (DTD) [3] has only 5,640 images and 47 texture labels. Our probing set uses a comparable number of images (64K) but yields >10× more unique visual concepts (16K), and is organized into a neuroscience-motivated hierarchy. So in the fine-grained interpretability literature, our scale is actually large.
> 2. Concept distribution matters for SAE interpretation. In Sec. 3.2, we explicitly study this. We fix the training image budget and train SAEs on our probing images, ImageNet, and MSCOCO. With identical architectures and hyperparameters, SAEs trained on our probing images achieve 3.4% higher interpretation accuracy (Tab. 3). Also, as mentioned above, compared to DN-CBM that trains SAE on CC3M, our steering outperforms DN-CBM by 34.4% accuracy on the worst-group Waterbirds. All this shows that carefully curated, abstraction-balanced data is more beneficial for interpretability than simply using an existing object-centric dataset.

---

> ### Author Response · Authors · 2025-11-25
> **Rebuttal by Authors (Part 2/2)**
>
> **Q3. Is the monosemanticity metric suited?**
> We agree that “monosemanticity” is a strong requirement and will soften the wording in the revision. Our “monosemanticity” score follows recent work [4], which evaluates monosemanticity by measuring how consistently the top-activating images for a feature share the same semantics. In our study, we use this metric as a **necessary but not sufficient indicator:** if a neuron fires on many semantically diverse images, it is clearly polysemantic; a high score suggests, but does not by itself guarantee, full disentanglement. Importantly, we do not base our conclusions on this metric alone. We **complement it with other quantitative criteria** to measure SAE behaviour on vision data (e.g., reconstruction quality, decoder orthogonality, dead-neuron rate), providing a comprehensive benchmarking of SAEs on vision data.
>
> &nbsp;
>
> **Q4. Paper presentation suggestions.**
> 1. **The X-ray vs MRI analogy is confusing.** We appreciate this point and will revise/remove the analogy.
> 2. **How to understand the naming of the Top-down concept reading / Bottom-up causal tracing?** In our work, “top-down” and “bottom-up” describe how the analysis is initiated. Following [5], “top-down” concept reading centers the analysis on representations and the transformations between them (e.g., sparse dictionary learning / SAE features) and then maps these representations to human concepts. “Bottom-up” causal tracing then follows the causal flow from earlier layers to later layers and the final prediction, by applying counterfactual edits and measuring their effect.
>
> &nbsp;
>
> **Q5. Implementation details.**
> 1. **How were nodes/edges in Fig. 5 selected?** Our graph pruning is straightforward. We first build the full concept-circuit graph (Sec. 2.3). For visualization, we rank concepts within each layer by their activation values and keep the top-k concepts per layer for visual simplicity. The sparsity/complexity of the circuit can be easily controlled by k. We will add these details (and exact hyperparameters) to Sec. 3.3 and Appendix F.2.
> 2. **How was the accuracy of GPT-5 annotations ensured?** We ensure the accuracy and reproducibility along three dimensions. (1) Careful prompt engineering (full prompt template in Appendix B) to ensure the LLM follows the instructions, generating consistent concept annotations. (2) Full details of the generation hyperparameters for the reproducibility, such as temperature=1.0, seed=42. (3) Additional human evaluation results of our concept set. As shown below, our concept set is faithful to the image and comprehensive across abstraction levels (>4.7/5 Likert score).
>
> | Evaluator    |     Human_A     |     Human_B     |     Human_C     | Avg. |
> |--------------|:---------------:|:---------------:|:---------------:|:----:|
> | Faithfulness | 4.65 $\pm$ 0.06 | 4.83 $\pm$ 0.20 | 4.90 $\pm$ 0.05 | 4.79 |
> | Completeness | 4.72 $\pm$ 0.04 | 4.73 $\pm$ 0.06 | 4.78 $\pm$ 0.18 | 4.74 |
>
> - Evaluation Data: Three groups of random images (3X50) paired with our concept annotations.
> - Metrics: Faithfulness (whether concepts are visually grounded in the image) and Comprehensiveness (whether they cover all visual abstraction levels).
> - Scoring: Both metrics are rated on a 0-5 Likert scale (0 = completely wrong, 5 = perfect match).
> - Evaluators: Three Computer Science major graduate students.
> - Conclusions: (1) The overall quality of our concepts is high. (2) Our concept annotation process is consistent across images.
>
> &nbsp;
>
> *We have incorporated all of these changes into the revised manuscript (highlighted in blue). We believe these clarifications and additional results address your concerns. We kindly ask the reviewer to take them into account when considering any score updates. We welcome any further questions or feedback.*
>
> &nbsp;
>
> [1] Rao et al. "Discover-then-name: Task-agnostic concept bottlenecks via automated concept discovery." ECCV’24.
> [2] Bau et al., Network Dissection: Quantifying Interpretability of Deep Visual Representations, CVPR 2017
> [3] Cimpoi et al. "Describing textures in the wild." CVPR’14.
> [4] Pach et al. "Sparse autoencoders learn monosemantic features in vision-language models." arXiv’25.
> [5] Zou et al. "Representation engineering: A top-down approach to AI transparency." arXiv’23.

---

> > ### Comment · Reviewer_CZUB · 2025-11-27
> >
> > Thank you for the comprehensive and thoughtful rebuttal, as well as the additional experiments and clarifications. However, even with these additions, some of my methodological and conceptual concerns remain. Please find my point-by-point response below:
> >
> > Q1. I appreciate the detailed comparison with DN-CBM and other SAE-based interpretability methods. The distinctions you draw are now much clearer. I think these differences should be made very explicit in the main paper, especially if the paper is accepted. In particular, spelling out how ViSAE is intended as an infrastructure/toolbox (rather than a new SAE variant) would be helpful.
> >
> > Q2. While I understand that your goal is probing a fixed ViT rather than re-pretraining a foundation model, I still have some reservations about how strong a prior your probing suite encodes. The motivation is neuroscience-inspired coverage of abstraction levels, but the overall philosophy of SAEs is to let the model surface its own internal concepts in a largely unsupervised way. Using a relatively confined, human-designed set of images and a concept vocabulary that is tightly grounded in our own prior abstractions move away from this spirit.
> >
> > Also, I am not fully convinced that 64K images are sufficient to argue that the resulting circuits reflect all relevant internal concepts, as opposed to those that align well with our prior taxonomy. I think it would help if the paper is more explicit about this limitation and about the implicit assumption that “good” ViT features aligned with human visual abstractions.
> >
> > Q3. I respectfully disagree that the metric used is even a sufficient indicator of monosemanticity. For example, how do we ensure that several images that contains object with similar object in a very different context are similar in visual representation space?
> >
> > Q4. Even after the rebuttal, I still find the “top-down concept reading” vs. “bottom-up causal tracing” terminology somewhat confusing. While "Bottom-up" refers to earlier vs later layers but this is not the case for the "top-down", which is defined more methodologically. I think those terms are confusing.
> >
> > Q5. Thank you for providing more details on the GPT-5 annotation process and the additional small-scale human evaluation. The scores suggest the concept annotations may be generally of good quality. However, given the small number of evaluators and the subjectivity of such ratings, I still see this more as a sanity check than as strong evidence of robustness. I agree it is still a useful addition to the paper, and it would be good to clearly frame this evaluation as a preliminary quality check rather than a gold-standard validation.
> >
> > Overall, the rebuttal improves the clarity and positioning of the work to some extent. However, the concerns above about the strength of the prior in the probing suite, the monosemanticity metric, and some of the methodological framing remain in my assessment.

---

> > > ### Author Response · Authors · 2025-11-29
> > > **Response to Reviewer CZUB’s Post-Rebuttal Comments**
> > >
> > > We thank the reviewer for **recognizing our novelty and contributions,** and for noting that the **"rebuttal improved the clarity and positioning"** of the work. Below, we address the remaining concerns point by point.
> > >
> > > **Q1. Positioning as "infrastructure/toolbox" and the explicit comparison with DN-CBM.**
> > > We appreciate the reviewer’s concrete suggestions. We have applied these suggestions in the revised version (Lines 58-66 and Lines 99-101), and will add more explicit comparisons in the related works.
> > >
> > > &nbsp;
> > >
> > > **Q2. Our probing suite design "move away from SAEs’ spirit".**
> > > We respectfully disagree. Our probing suite is designed to **support,** not constrain, the SAE’s ability to "surface internal concepts in a largely unsupervised way". The SAE training itself **remains fully unsupervised** on ViT representations. What we improve is the **quality of the playground** it learns from. Compared to off-the-shelf datasets like ImageNet, our probing data offers more diverse and comprehensive concept coverage, giving SAEs more (not fewer) opportunities to reveal meaningful features. We clarify this through two subquestions:
> > >
> > > **(1) Probing image set: Does simply scaling up probing images help SAE interpretations?**
> > > Not necessarily. Scaling helps when it can actually increases concept coverage.
> > > - **With strategy (ours):** Neuroscience-motivated design enlarges the coverage of visual abstractions, which improves what SAEs can discover.
> > > - **Without strategy (naive scaling):** Blindly adding more homogeneous data provides limited contributions to concept coverage while introducing significant computation overhead (Tab. 1).
> > >
> > > This is supported both theoretically and empirically:
> > > **(i) Theoretically,** SAEs are not magic black boxes that can "interpret everything". Prior work [1] shows that due to the duality between SAEs and concept geometry, what SAEs can interpret is largely shaped by their architecture and the training distribution, not simply by data volume.
> > > **(ii) Empirically,** we compare SAEs trained on our 64K probing images vs. SAEs trained on the full 1.28M ImageNet. As shown in the table below, naive scaling (ImageNet) does not improve interpretation accuracy or downstream steering, whereas our probing images outperform ImageNet by 2.6% and 30.1%, respectively.
> > > | Probing Image Set | \# of Images | Interp. Acc. (Top-30) | Worst-group Acc. (waterbird) |
> > > |---|:---:|:---:|---|
> > > | ImageNet (down-sample) | 64K | 50.7 | 63.9 |
> > > | ImageNet (full) | 1,280K | 51.5 | 71.4 |
> > > | Ours | 64K | **54.1** | **98.5** |
> > >
> > > **(2) Concept set: Do we implicitly assume that "good" ViT features are those aligned with human visual abstractions?**
> > > No. We do **not** assume what kind of ViT feature is "good". Instead, we empirically study **what concept sets are good for interpreting ViT features.** Our finding is that **visually grounded concepts** are more effective than **linguistically frequent concepts** for interpreting a vision model: (i) ViTs are trained on images, so the concepts they encode are primarily grounded in visual patterns, not in word frequency. (ii) In Tab. 3, our concept set, constructed from image content, outperforms existing concept vocabularies mined from frequent English words or LAION captions by 28.7% and 37.4% in interpretation accuracy.
> > >
> > > &nbsp;
> > >
> > > **Q3. Presentation suggestions.**
> > > **Is monosemanticity a sufficient indicator?** No. In the previous rebuttal, we already agreed with the reviewer that this metric is a necessary but **not sufficient** indicator for monosemanticity. Note that this metric is **not** our contribution, we directly adopt it from [2] as one of the metrics to explore the best SAE architecture for the vision. **We complement it with many other quantitative criteria** to measure SAE performance on vision data (e.g., reconstruction quality, decoder orthogonality, dead-neuron rate), not solely rely on this metric alone. We will highlight the limitation of this metric in the revised version.
> > >
> > > **"Top-down" and "Bottom-up" naming suggestions.** We thank the reviewer’s feedback on the terminology. We will adjust them to more direct names, e.g., "concept reading" and "concept circuit tracing".
> > >
> > > &nbsp;
> > >
> > > **Q4. Can our human evaluations validate the quality of our concept set? Can this be scaled up?**
> > > We appreciate the reviewer’s acknowledgement that the annotations appear **generally high-quality,** and we agree that our current human evaluation is a sanity check of our concept set. Regarding scaling up, a full-scale, gold-standard validation of the entire concept set is beyond the scope of this rebuttal period, but we see it as a natural direction for follow-up work and will mention this explicitly in the revised version.
> > >
> > > &nbsp;
> > >
> > > [1] Hindupur et al. "Projecting assumptions: The duality between sparse autoencoders and concept geometry." ICML’25W.
> > > [2] Pach et al. "Sparse autoencoders learn monosemantic features in vision-language models." arXiv'25.

---

### Author Response · Authors · 2025-11-25
**Rebuttal by Authors (Part 1/2)**

Dear Reviewers, Area Chairs, and Program Chairs,

Thank you for your time and effort in reviewing our paper. We appreciate the reviewers find our idea `"addressed the important problem of AI safety"`(CZUB), `"Tackles a core gap in ViT interpretability"` (q9sp), `"significantly helps ViT interpretability"` and `"bring practical benefits"` (DS6U); Our toolbox is `"a good addition to the interpretability arsenal"` (q9sp), `"has many interesting components"` and `"would be a great resource for future research"` (CZUB); Our method has `"clean, modular pipeline"` and `"providing a computation path that's interpretable"` (q9sp); Our experiments perform `"An extensive benchmarking of different types of SAE for vision data"` (D6zH); Our paper is `"well written and is easy to read"` (DS6U), with `"nice figures"` and `"a clear list of contributions"`. We have responded to the individual comments of each reviewer.

**We summarize the main comments of the four reviewers below:**

1. **The major concern of Reviewers CZUB and D6zH is that “the technical novelty is limited”.**
We respectfully disagree. The reviewers’ novelty concerns mainly target SAE architecture or optimization, whereas our contribution is different in nature. Our novelty is **not** another SAE variant. Instead, we introduce **a toolbox for interpreting the inner workings of ViTs:** a faithful **infrastructure** that is largely **missing** in the existing literature. The toolbox consists of: **(i) a neuroscience-motivated probing suite** of 64K images and 16K visual concepts, **(ii)** a **top-down** concept reading algorithm, and **(iii)** a **bottom-up** concept circuit tracing algorithm. Here, SAEs serve as one component within the concept-reading stage, rather than the main contribution. Building such a toolbox is non-trivial, and we address two concrete gaps:

**(1) What probing data enables faithful interpretation of ViT inner workings?** Recent studies [1] show that what SAEs can see and explain is shaped by the data they are trained on and interpret. However, existing vision SAE works [2,3,4] often rely on off-the-shelf image sets (e.g., ImageNet) and generic text vocabularies. Consequently, their interpretations are typically skewed toward object-level concepts (Lines 65-70). To address this, motivated by the **human visual cortex hierarchy** from neuroscience [5], we construct a **first-of-its-kind probing suite** (64K images + 16K concepts) that ensures SAEs learn concepts spanning the full spectrum of visual processing. It covers concepts from low-level primitives (e.g., colors and edges), through intermediate textures and shapes, to high-level objects and parts, and finally scene-level motions and relations. Empirically, our probing images achieve **20× more efficient** concept coverage (table below), and our concept set enables **auto-interpretation** and improves its accuracy by 28.7% over existing vocabularies (Tab. 3, rows 3-5).

| Probing Image Set | # of Images | Concepts Covered by Images (%) |  |  |  |  | Coverage Efficiency $\uparrow$ (%/1K Images) |
|---|---:|---:|---:|---:|---:|---:|---:|
|  |  | Primitive | Intermediate | Object | Scene | Avg. |  |
| ImageNet | 1,281K | 81.0 | 78.2 | 97.7 | 59.0 | 78.9 | 0.06 |
| MSCOCO | 118K | 69.6 | 65.4 | 80.4 | 63.1 | 69.6 | 0.59 |
| Ours | 64K | 87.1 | 80.6 | 92.6 | 61.7 | **80.5** | **1.26** |

**(2) How to represent the inner workings of ViT in a human-readable way?** Existing vision SAE works typically extract **discrete concepts from a single layer,** often the final embeddings (Lines 70–74). As a result, they miss a key aspect of ViT inner workings: how concepts interact and propagate across layers. To fill this gap, our toolbox provides (i) a top-down step that reads human concepts from ViT representations, and (ii) a bottom-up step that traces their cross-layer interactions. Together, our toolbox offers **structured cross-layer concept circuits** that capture how concepts emerge, evolve, and interact to jointly drive predictions. This mechanism view significantly benefits downstream steering, improving worst-group accuracy by 33.7% (Tab. 4).

---

> ### Author Response · Authors · 2025-11-25
> **Rebuttal by Authors (Part 2/2)**
>
> 2. **Reviewers CZUB and q9sp are concerned about the “accuracy of GPT-5 automatic concept annotations”.**
> We ensure the accuracy and reproducibility along three dimensions. (1) Careful prompt engineering (full prompt template in Appendix B) to ensure the LLM follows the instructions, generating consistent concept annotations. (2) Full details of the generation hyperparameters for the reproducibility, such as temperature=1.0, seed=42. (3) Additional human evaluation results of our concept set. As shown below, our concept set is faithful to the image and comprehensive across abstraction levels, proven by a >4.7/5 Likert score.
>
> | Evaluator    |     Human_A     |     Human_B     |     Human_C     | Avg. |
> |--------------|:---------------:|:---------------:|:---------------:|:----:|
> | Faithfulness | 4.65 $\pm$ 0.06 | 4.83 $\pm$ 0.20 | 4.90 $\pm$ 0.05 | 4.79 |
> | Completeness | 4.72 $\pm$ 0.04 | 4.73 $\pm$ 0.06 | 4.78 $\pm$ 0.18 | 4.74 |
>
> - Evaluation Data: Three groups of random images (3X50) paired with our concept annotations.
> - Metrics: Faithfulness (whether concepts are visually grounded in the image) and Comprehensiveness (whether they cover all visual abstraction levels).
> - Scoring: Both metrics are rated on a 0-5 Likert scale (0 = completely wrong, 5 = perfect match).
> - Evaluators: Three Computer Science major graduate students.
> - Conclusions: (1) The overall quality of our concepts is high. (2) Our concept annotation process is consistent across images.
>
>
> &nbsp;
>
> 3. **Reviewer D6zH requests additional comparisons.**
> While we believe our existing results already demonstrate the strength of our toolbox, we agree that adding more comparisons further clarifies its advantages. We have therefore included the following new results:
> (1) SAE training and interpretation: Our probing image set and concept set outperform Broden [6] and LaBo [7] by 27.3% and 33.8% on interpretation accuracy.
>
> |                   |               | Interpretation Accuracy (%) |          |          |
> |-------------------|---------------|--------------------------|----------|----------|
> | Probing Image Set |  Concept Set  |          Top-10          |  Top-20  |  Top-30  |
> |       Broden      |     Broden    |           16.8           |   23.9   |   26.8   |
> |      ImageNet     | LaBo-ImageNet |           15.9           |   18.4   |   20.3   |
> |        Ours       |      Ours     |         **36.6**         | **47.6** | **54.1** |
>
> (2) Concept localization: Our heatmaps on the Quantus [8] benchmark improve over the existing attribution-based method [9] by 3.7% using the VOC2007 dataset.
> | Method        | Point Game | Attribution Localization |
> |---------------|:----------:|:------------------------:|
> | Chefer et al. |    41.9    |           32.3           |
> | Ours          |  **45.0**  |         **36.0**         |
>
> (3) Model steering (Waterbirds dataset): Our method outperforms post-hoc CBM [10] and DN-CBM [11] by 23.8% and 34.4% on worst-group accuracy.
> | Method | CBM | SpLiCE | DN-CBM | PCBM | Ours |
> |---|---|---|---|---|---|
> | Worst Group Acc. (before steer) | 37.3 | 48.0 | 57.5 | 50.3 | 50.3 |
> | Worst Group Acc. (after steer) | 51.8 | 60.0 | 71.3 | 74.7 | **98.5** |
> | $\Delta$ | +14.5 | +12.0 | +13.8 | +24.4 | **+48.2** |
>
> &nbsp;
>
> *We have incorporated all of these changes into the revised manuscript (highlighted in blue). We believe these clarifications and additional results address the reviewers’ concerns, and we kindly ask the reviewers to take them into account when considering any score updates. We welcome any further questions or feedback from the reviewers.*
>
> &nbsp;
>
> [1] Hindupur et al. "Projecting assumptions: The duality between sparse autoencoders and concept geometry." ICML’25W.
> [2] Lim et al. "Sparse autoencoders reveal selective remapping of visual concepts during adaptation." ICLR’25.
> [3] Thasarathan et al. "Universal sparse autoencoders: Interpretable cross-model concept alignment." ICML’25.
> [4] Rao et al. "Discover-then-name: Task-agnostic concept bottlenecks via automated concept discovery." ECCV’24.
> [5] Goodale et al. "Separate visual pathways for perception and action." Trends in Neurosciences, 1992.
> [6] Bau et al., Network Dissection: Quantifying Interpretability of Deep Visual Representations, CVPR’17.
> [7] Yang et al., Language in a Bottle: Language Model Guided Concept Bottlenecks for Interpretable Image Classification, CVPR’23.
> [8] Hedström et al., Quantus: An Explainable AI Toolkit for Responsible Evaluation of Neural Network Explanations and Beyond, JMLR’23.
> [9] Chefer et al. "Generic attention-model explainability for interpreting bi-modal and encoder-decoder transformers." ICCV’21.
> [10] Yuksekgonul et al., Post-hoc Concept Bottleneck Models, ICLR’23.
> [11] Rao et al. "Discover-then-name: Task-agnostic concept bottlenecks via automated concept discovery." ECCV’24.

---

### Meta-Review · Area_Chair_HNgS · 2026-01-03

**Summary:**

Across all reviewers, the central concerns are:

a) The paper’s technical novelty and methodological contribution are limited, with overlap with prior work (e.g., CLIP-Dissect, network dissection, causal tracing, concept steering, and recent SAE-based interpretability papers) that is either insufficiently cited or not clearly differentiated.

b) The proposed probing dataset and GPT-5 – generated concept annotations, while useful, are viewed as incremental and weakly validated, with concerns about bias, reproducibility, and circular evaluation. Reviewers also question the adequacy of the probing image set size (64K) for training SAEs and discovering comprehensive internal concepts, as well as the validity of the monosemanticity metric, which may conflate feature consistency with true conceptual disentanglement.

c) Experimentally, reviewers find certain baselines are outdated or incomplete (missing newer CBM variants and SAE-based methods). Quantitative evaluation of visualizations (e.g. Fig. 6) is lacking, with unexplained inconsistencies.

**Reviewer Concerns:**

In the rebuttal, the authors provided additional experimental results on SAE training and interpretation, concept localization, and model steering (including experiments on the Waterbirds dataset), and added comparisons to more recent methods such as Network Dissection and Language in a Bottle. These additions help demonstrate the practical usefulness of the proposed toolbox and partially strengthen the empirical evidence.

However, several concerns remain insufficiently addressed. (a) Although the authors clarified their positioning relative to prior work (e.g., DN-CBM, ECCV’24) and elaborated on their technical contributions, extending existing SAE analyses from the final embedding layer to the full ViT architecture still appears incremental. (b) While a small-scale human evaluation was presented to assess the accuracy and extensiveness of GPT-5 – generated concept annotations, the limited scale and subjective nature of this study make it more of a sanity check than a convincing validation. Relatedly, it remains debatable whether the proposed dataset scale (64K images with concepts) is sufficient to capture a broad and representative set of internal model concepts. (c) Finally, technical contributions from the algorithm design perspective and several inconsistencies observed in the visualizations (raised by Reviewer D6zH) remain unexplained.

**Reviewer Scores:**

(a) Reviewer CZUB actively participated in the discussion to a substantial extent; their final score would either remain at 2 or be increased to 4.
(b) Reviewer q9sp would maintain their original score, given the low confidence expressed in the initial review.
(c) Reviewer DS6U would maintain their original score.
(d) Reviewer D6zH would either maintain a score of 2 or raise it to 4.

---

### Decision · Program_Chairs · 2026-01-26

Reject